# Sex-specific developmental gene expression atlas unveils dimorphic gene networks in *C. elegans*

Rizwanul Haque [1,2], Sonu Peedikayil Kurien[1,2], Hagar Setty [1,2], Yehuda Salzberg[1,2], Gil Stelzer [3], Einav Litvak[1], Hila Gingold[4], Oded Rechavi[4] & Meital Oren-Suissa [1,2] ✉

Sex-specific traits and behaviors emerge during development by the acquisition of unique properties in the nervous system of each sex. However, the genetic events responsible for introducing these sex-specific features remain poorly understood. In this study, we create a comprehensive gene expression atlas of pure populations of hermaphrodites and males of the nematode *Caenorhabditis elegans* across development. We discover numerous differentially expressed genes, including neuronal gene families like transcription factors, neuropeptides, and G protein-coupled receptors. We identify INS-39, an insulin-like peptide, as a prominent male-biased gene expressed specifically in ciliated sensory neurons. We show that INS-39 serves as an early-stage male marker, facilitating the effective isolation of males in high-throughput experiments. Through complex and sex-specific regulation, *ins-39* plays pleiotropic sexually dimorphic roles in various behaviors, while also playing a shared, dimorphic role in early life stress. This study offers a comparative sexual and developmental gene expression database for *C. elegans*. Furthermore, it highlights conserved genes that may underlie the sexually dimorphic manifestation of different human diseases.

Genetic sex introduces variation in phenotypic traits in sexually reproducing organisms. Sexual dimorphism, the phenotypic differences between the two sexes of a species, can be exhibited at various levels. These differences include variances in morphology[1], sensory sensitivity[2], social behavior[3,4], and disease progression or onset[5]. In essentially all sexually reproducing animals, from nematodes to humans, the nervous system undergoes sexual differentiation during a crucial developmental period, leading to changes in neuroanatomy, neural differentiation, synaptic connectivity, and physiology, which profoundly influence behavior[6–10]. This developmental shift is driven and accompanied by a sex-specific transcriptional program that has only been partially studied. Past work on developmental transcriptomes of various organisms, while capturing some universal

features of sexual differentiation, were usually limited in scope, either focusing on one sex only[11,12] or ignoring sex altogether[13,14], covering limited developmental timepoints[15], or focusing on non-neuronal tissues[11,12,16].

To gain insight into the molecular and genetic mechanisms underlying the sexual component of nervous system development, we asked how sexual identity, neuronal identity, and developmental stage intersect to drive gene expression in a model organism. We addressed this question using the nematode *Caenorhabditis elegans* (*C. elegans*), due to the detailed anatomical and molecular understanding of the nervous system of both sexes and the extensive sexual dimorphism they exhibit[17,18] at the resolution of single identifiable neurons, connections, and behaviors[2,19–21]. As most sexual differences arise late in

[1]Department of Brain Sciences, Weizmann Institute of Science, Rehovot, Israel. [2]Department of Molecular Neuroscience, Weizmann Institute of Science, Rehovot, Israel. [3]Department of Life Sciences Core Facilities, Weizmann Institute of Science, Rehovot, Israel. [4]Department of Neurobiology, Wise Faculty of Life Sciences & Sagol School of Neuroscience, Tel Aviv University, Tel Aviv, Israel. ✉e-mail: meital.oren@weizmann.ac.il

development, *C. elegans* offers a unique opportunity to track how sex-specific characteristics emerge during neuronal development.

*C. elegans* is an androdioecious nematode, with hermaphrodites (XX) and males (XO). Males are rare (0.01%) in the standard lab strain, and no obvious morphological or molecular markers exist for large-scale male isolation before sexual maturation (stretching roughly from the late L3 stage up to the transition into adulthood), a long-standing obstacle for systemic studies of sexual dimorphism or male development. Therefore, previous genome-wide expression studies in *C. elegans* have generally centered around just one sex, the hermaphrodite, and were limited to late larval stages or the use of pseudo males and relatively few stage-specific sample replicates[22–27]. Traditional methods to purify adult males rely on manual picking[28], mutations in *him* (*h*igh *i*ncidence of *m*ales)[29] genes, or by size exclusion using mesh filters[30], all of which are labor intensive and time-consuming. In a recent study[31], large-scale preparation of L4 male larvae was achieved using auxin-induced degradation of a *d*osage *c*ompensation *c*omplex (DCC) component; nevertheless, male isolation at the early larval stages is still absent.

In the current work, we describe a gene expression atlas for the two sexes of *C. elegans* across development (https://www.weizmann.ac.il/dimorgena/). To achieve this, we developed a methodology that enables large-scale isolation of early larval males with high purity and then carried out whole animal RNA sequencing across all significant developmental stages for both sexes. Our findings reveal a multitude of sexually dimorphic differentially expressed genes, including neuronal gene families such as transcription factors (including DM domain and homeobox genes), neuropeptides, and G protein-coupled receptors (GPCRs). We validated our findings using multiple approaches and identified an early-stage marker for males. Comprehensive anatomical localization and functional studies of one of the top hit genes, the insulin-like neuropeptide INS-39, revealed sexually dimorphic expression and functions in the two sexes. We also discovered a complex and sexually dimorphic regulation system for this gene. Together, the extensive database of dimorphic gene expression across development provided by this study will serve as a source for determining the functions of specific genes in both sexes. Finally, our study also draws attention to conserved candidate genes that might be linked to the sexually dimorphic manifestations of human diseases.

## Results

### Large-scale male isolation in *C. elegans* L1 larvae
To obtain a male-enriched worm population we exploited a temperature-sensitive mutation in a dosage compensation complex (DCC) gene detrimental only to hermaphrodites. The DCC, a specialized regulatory mechanism that downregulates gene expression from the two X chromosomes, is only formed in *C. elegans* hermaphrodites (males carry just one X chromosome) (Fig. 1a)[32]. Consequently, defective DCC will result in a lethal overdose of gene expression from the X chromosomes only in hermaphrodites[33–35]. We used a temperature-sensitive allele (*y1*) in the DCC gene *dpy-28*[36] combined with a *him-8* mutation that spontaneously leads to a high incidence of males (Fig. 1b). As expected, at the restrictive temperature (25 °C), most XX hermaphrodites died as embryos or L1 larvae, whereas the XO males were unaffected. However, even at the restrictive temperature, 20% of XX hermaphrodites were still viable (Fig. 1b). We observed that the L1 XX hermaphrodites that did hatch at the restrictive temperature were severely impaired, displaying body defects and defective locomotion. We utilized this sex-specific phenotype and improved male purity by slicing out and removing the immobile drop of hermaphrodites from the plate (Supplementary Fig. 1, see methods). Using this modified protocol, we improved male enrichment to over 98% (Fig. 1b).

To determine the suitability of the isolated *dpy-28* males to serve as a source for wild-type male RNA sequencing, we assayed their morphology, locomotion, and mating behavior, and compared them to wild-type males. *dpy-28* males were indistinguishable from *him-8* control males in multiple parameters and behaviors, including body size, locomotion, and mating efficiency (Supplementary Fig. 2). Relatedly, previous studies have shown that RNAi against *dpy-28* exclusively affects hermaphrodites, and not males, in lifespan and dauer arrest[37,38]. Therefore, we concluded that, unlike *dpy-28* hermaphrodites, *dpy-28* males develop normally into healthy adults, and can thus serve as an opposite-sex counterpart to wild-type hermaphrodites for comparative transcriptome analyses.

### Genetic sex and development shape animal transcriptomes
Having established a reliable procedure to obtain pure male populations starting from early juvenile stages, we sought to carry out whole-animal RNA-seq transcriptomics in both sexes and throughout all developmental stages. To achieve this, we used a modified bulk version of the single-cell MARS-seq protocol. This modified approach is a high-throughput, low-input 3′-mRNA-seq method, which enhances the quality of library preparation for more accurate gene expression profiling[39,40]. We produced RNA-seq profiles from five distinct developmental stages (L1, L2, L3, L4, YA) for both sexes, totaling 40 samples (10 groups with four biological repeats) (Fig. 1c). Male percentages, raw read counts, and RINe scores per sample are presented in Supplementary Data 1. PCA analysis and hierarchical clustering of pairwise sample Pearson correlations grouped RNA profiles from biological repeats with high confidence, exhibiting the highest correlations for intra-group samples (Fig. 1d, e). Notably, when our samples were separated by stage on principal component 1 (PC1), and by sex on principal component 3 (PC3), samples from both sexes clustered together before the onset of sexual maturation (i.e., L1-L3 stages), and robustly diverged after sexual maturation (L4-YA stages), reflecting the dramatic anatomical and physiological changes that occur during sexual maturation. We detected expression data for 14185 genes (Fig. 1f), among which 4698 genes (33%) showed dimorphic expression in either developmental stage, including 2474 known genes and 2224 predicted novel genes. The expression data revealed 519 (L1), 63 (L2), 614 (L3), 2141 (L4), and 3564 (YA) genes that were differentially or dimorphically expressed (Fig. 1g, for raw expression data see Supplementary Data 2). Overall, there were more male-biased differentially expressed genes (DEGs) (henceforth called male enriched genes (MEGs)) than hermaphrodite-biased DEGs (hermaphrodite enriched genes (HEGs)) at all developmental stages, suggesting an inherent sex-dependent bias of gene expression (Supplementary Fig. 3a).

To validate our datasets, we confirmed the expected expression pattern of known markers necessary for the development of sex-specific features, which typically appear during the third larval stage[41]. *pkd-2*, for instance, is expressed in the cilia of three different types of male sensory neurons[42], *K09C8.2* in the seminal vesicle and vas deferens[43], *clec-207* in the vas deferens[26], and *Y39B6A.9* in male spicule muscle[27], *R01E6.5* in male spicule neuron[27] and *ram-5* in the ray neurons[44,45]. For hermaphrodites, we used *meg-1*, which is expressed in the proximal germline[46], and *vit-2* in the vesicles of the oocyte[47]. Our data coincides with these genes' reported stage-specific expression pattern, which shows essentially no to minimal expression in males/hermaphrodites before sexual maturation and increases drastically during L4 when sex-specific structures emerge (Supplementary Fig. 3c–j).

Next, we sought to compare DEGs between stages. We analyzed our datasets using jvenn[48] and discovered that only 10 male-enriched genes and 3 hermaphrodite-enriched genes remained enriched sex-specifically throughout all developmental stages (Fig. 1g, Supplementary Data 3). Expectedly, the highest number of DEGs were found at the L4-Adult stages, in both sexes. We compared our sex-enriched genes from the L3 to YA stage with a previous publication by Kim et al. that reported male-specific and nervous-system-specific expression of

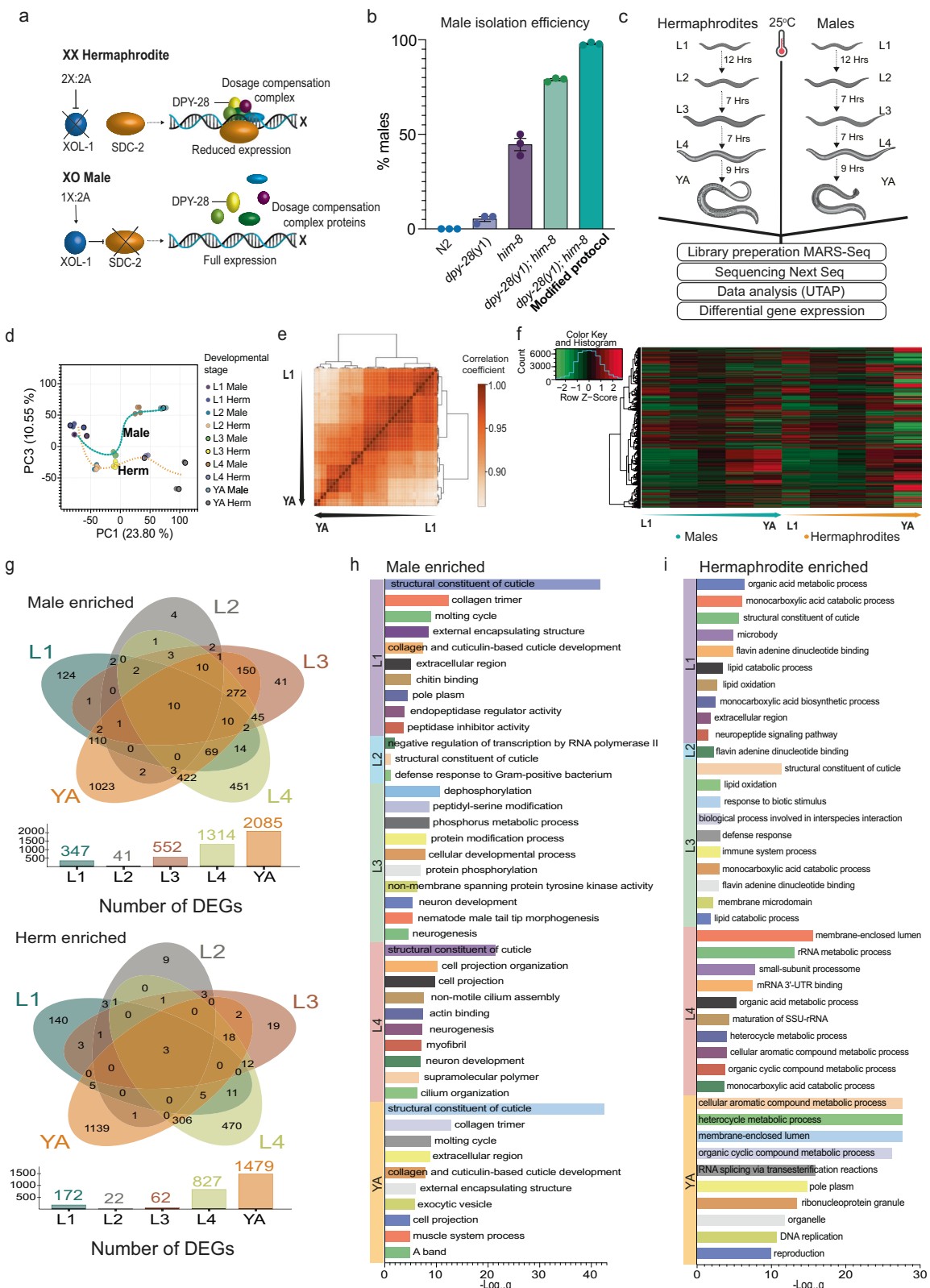

genes after sexual maturation and found an overlap of many genes (Supplementary Data 4, Supplementary Fig. 4). Interestingly, our analysis uncovered twice as many DEGs in males at the L1 stage compared with L1 hermaphrodites (Supplementary Fig. 3a). The cellular anatomy of L1 males and hermaphrodites is similar, and only a few differences have been reported, such as the size difference of the

epithelial B cell[49]. Our analysis uncovered previously undescribed early male-biased expression of a heterochronic gene, *lin-42*, which regulates temporal cell identities[50,51] (Supplementary Fig. 3k). Another example is the Zn finger transcription factor *lin-29* which has been shown to regulate timing mechanisms of sexually dimorphic nervous system differentiation[52] (Supplementary Fig. 3l). Our results suggest

**Fig. 1 | Large-scale isolation of pure male populations enables profiling of whole animal transcriptomes across all developmental stages. a** A model for *C. elegans* dosage compensation complex (DCC) forming on the X chromosome only in hermaphrodites (image modified with permission from[129]). Created with Biorender.com. **b** Temperature-sensitive mutation (*y1*) in the dosage compensation gene, *dpy-28*, facilitates large-scale male isolation. Quantification of male percent at YA developmental stage in N2, *dpy-28(y1)*, *him-8(e1489)*, *dpy-28(y1);him-8(e1489)* and *dpy-28(y1);him-8(e1489)* (modified protocol) at 25ºC with 3 biological replicates, n = 100 worms per group. Data are presented as mean values +/- SEM. **c** Schematic of experimental design and RNA-seq procedure (see methods). Created with Biorender.com. **d** Principal component analysis projection of expression patterns for the five developmental stages of the two sexes. Each circle represents individual samples. The progression of developmental stages is shown as a dotted line for each sex (cyan, males, and orange, hermaphrodites). The % variance, out of the total original variance in the high-dimensional space, spanned by the first and third PCs is indicated on the x and y axis, respectively. **e** Sample correlations.

Hierarchical clustering of all sample types based on their RNA expression profiles. Off-diagonal entries denote the Pearson correlation between the expression profiles of two different sample types. Here, the sample classes cluster well, showing the highest intra-group correlations (developmental stage and sex). **f** Hierarchical clustering of differentially expressed genes (by DESeq2) using the genes expression values (rlog transformed counts (rld)) on a per-developmental stage basis, using the thresholds for significant differential expression as padj ≤ 0.05, |log$_2$ fold change|>= 1 and basemean >= 5. Each row represents a gene. The arrow indicates the progress of developmental stages and results are clustered only by rows. **g** Venn diagrams showing genes enriched in males (top) and hermaphrodites (bottom) across all developmental stages. Top 10 GO term enrichments for males enriched (**h**) and hermaphrodites enriched (**i**) set of genes for each developmental stage. DEGs, differentially expressed genes. The significance of the enrichment at a particular stage was determined using *q*-value threshold of 0.1. Source data are provided as a Source Data file.

that its differential expression is set up earlier than previously reported. As comparisons of the sexes at the ultrastructural levels haven't been made at these early larval stages, we lack knowledge of the extent of dimorphic wiring established early during development. Some of the identified genes might shed light on these wiring events (Supplementary Data 5). To gain further insight, WormBase gene ontology (GO) enrichment analyses were applied to the DEGs for each developmental stage and sex[53]. Male-enriched genes at early development were significantly associated with cuticle development and at later stages with neurogenesis and molting (Fig. 1h). In contrast, hermaphrodite-enriched genes at early development were significantly associated with metabolic pathways and at later stages with reproduction and gamete formation (Fig. 1i). Taken together, profiling of whole animal transcriptomes across all developmental stages reveals a distinct sex-specific developmental plan before any morphological sexual features arise.

## Sexually dimorphic regulation of neuronal gene families

From worms to humans, the nervous system undergoes extensive transcriptional and functional transformation upon sexual maturation[18,21,54]. We thus searched specifically for neuronal gene families that display sexually dimorphic expression in our dataset. We examined transcription factor families[55], including DM domain genes[56] and homeobox genes. 218 transcription factors (TFs) were differentially expressed in at least one developmental stage (Supplementary Data 6, Top 25 represented in Fig. 2a), including previously known male-specific TFs[45,57] such as *mab-3*, *mab-5*, *egl-5*, *dmd-3* and the hermaphrodite-specific TFs[58,59] *lin-13* and *lin-15*. This list uncovered a pool of TFs with temporal and sexual specificity during development. While only a few TFs were differentially expressed at early development (Supplementary Fig. 3b), we found over 100 that were differentially expressed in young adults (Supplementary Data 6). 73 TFs were specifically enriched in males, and 74 were specifically enriched in hermaphrodites. Intriguingly, we found just two TFs in our data, *ces-2* and *mab-3*, that were consistently higher in males across all developmental stages (Fig. 2a–d, Supplementary Fig. 5e). The DM (**D**oublesex/**M**AB-3) domain (DMDs) genes regulate sexual development across evolution and are integral players in sexual development and its evolution in many metazoans[60]. We found three DM domain genes to be expressed dimorphically, namely *mab-3*, *dmd-4*, and *dmd-3* (Fig. 2d and Supplementary Data 6), all of which have been previously shown to play sex-specific roles[45,61,62]. We also noticed that most DMDs are regulated temporally in both sexes, suggesting they might play stage-specific roles in shared developmental processes.

Homeobox genes regulate various aspects of development and specification of neuronal identity in *C. elegans* and across evolution[63]. It was recently shown that neuronal diversity in *C. elegans* can be fully described by unique combinations of the expression of homeobox

genes[64–66]. Thus, dimorphism in homeobox gene expression might imply that there is sexual dimorphism also in the code that defines the neuronal identity features. We found that 19 of the homeobox-containing genes were sexually dimorphic across various stages of development (Fig. 2e, Supplementary Data 6). Our data corroborate the high male expression of *egl-5*, *vab-3*, *mab-5*, *ceh-13*, and *nob-1*, required for several aspects of male sexual differentiation, like the formation of male-specific sensory organs, sex muscle differentiation, or gonadal development[67–71]. Additionally, we found that *ttx-1*, *zfh-2*, and *zag-1*, previously shown to be involved in neurogenesis[72–74] and *irx-1* in synapse elimination[75], are expressed significantly higher in males compared to hermaphrodites during sexual maturation (Fig. 2e) and can thus be potentially involved in male-specific neurogenesis/rewiring which have not been explored before. Similarly, we found extensive sexual dimorphism throughout development for neuronal terminal differentiation genes, such as K channels, ligand-gated ion channels, ionotropic receptors, synaptic vesicle genes, and nuclear hormone receptors[76–78] (Supplementary Fig. 6a–e, Supplementary Data 6). In summary, our data provide a rich resource for the factors that may drive male neuronal identity and its functional landscape.

## The insulin-like peptide INS-39 is highly sexually dimorphic

Neuropeptides constitute a vast class of signaling molecules in the nervous system of many groups of animals, yet despite their prevalence, dimorphic functions have been assigned only to a few neuropeptides[79–85]. We thus focused on the large, extended family of neuropeptides in *C. elegans* and their cognate GPCR receptors. Among the receptors, *srj-49* caught our attention for being extremely dimorphic as it is upregulated in males starting already at the L1 stage and throughout all subsequent developmental stages (Fig. 3a). Real-time qPCR analysis validated that *srj-49* mRNA is absent in hermaphrodites at all stages, while it is highly expressed in males (Supplementary Fig. 5a, b). However, we observed no detectable SRJ-49 protein neither by using an *srj-49*p::GFP multi-copy fosmid reporter nor by a single-copy *srj-49*::SL2::GFP::H2B CRISPR reporter (Supplementary Fig. 5c, d). *srj-49* could potentially be regulated post-translationally in a sex specific-manner, as we have previously demonstrated for a synaptic receptor[86].

The *C. elegans* genome encodes 154 known neuropeptide genes, 40 genes belong to the insulin-like family of peptides, 31 genes are FMRFamide-related peptides, and 83 genes encode non-insulin, non-FMRFamide-related neuropeptides[87]. Our analysis shows that 37 neuropeptide genes exhibit dimorphic expression with a strong bias toward male enrichment (Fig. 3b). We found that only 12 neuropeptide genes exhibited dimorphic expression before the L4 stage, indicating that most neuropeptide-dependent signaling pathways diverge only after sexual differentiation. Interestingly, we found a

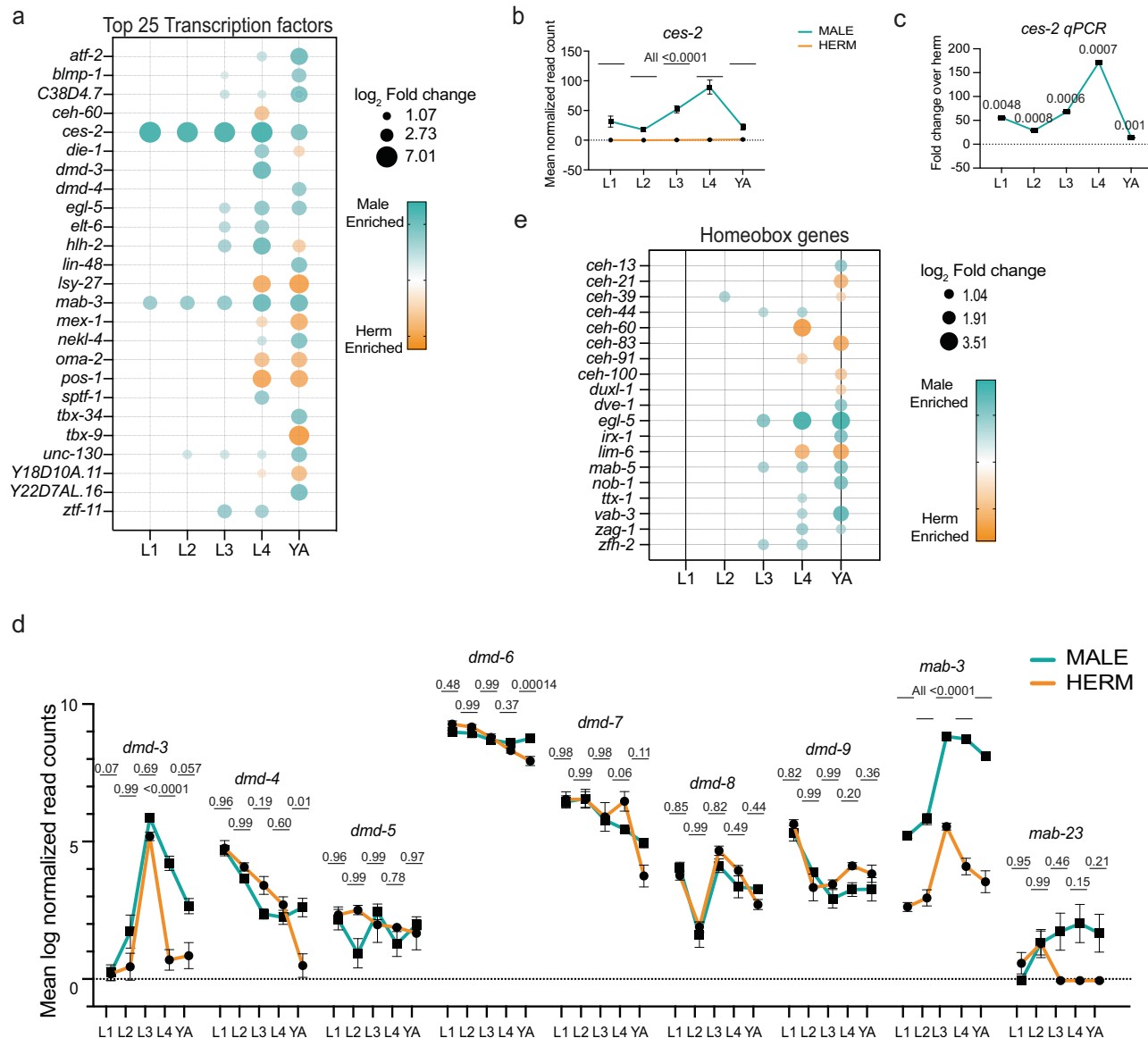

**Fig. 2 | Selected gene families with sexually dimorphic regulation. a** Bubble plot representation of the top 25 transcription factors out of 218 differentially expressed in any of the five developmental stages of the two sexes. Bubble size represents log$_2$ of fold change in expression of that gene (only genes that passed the filter padj <= 0.05, |log$_2$ fold change|>= 1 and base mean >= 5 are plotted), bubble color (cyan: males and orange: hermaphrodites) represents enrichment in either sex. TFs were mined using wTF2.2, an updated compendium of *C. elegans* transcription factors[55]. **b** Normalized RNA-seq expression values of *ces-2* across all developmental stages, n = 4 biological repeats per sample. *P*-values were <0.0001 for all comparisons between sexes in all developmental stages **c** Real-time qPCR analysis of *ces-2* mRNA (*ces-2* mRNA male expression normalized to hermaphrodite expression) across all developmental stages, n = 3 biological repeats per sample. **d** Log$_2$ normalized expression values of differentially expressed DM domain genes across development, n = 4 biological repeats per sample. **e** Bubble plot representation of homeobox genes differentially expressed in any of the five developmental stages of the two sexes. Bubble size represents log$_2$ of fold change in expression of that gene (only genes that passed the filter padj <= 0.05, |log$_2$ fold change|>= 1 and base mean >= 5 are plotted), bubble color (cyan: males and orange: hermaphrodites) represents enrichment in either sex. In **b**, **c**, and **d** data are presented as mean values +/− SEM. In c *p*-values were calculated by a two-sided t-test for each comparison performed. In **b**, **d** adjusted p-values were calculated by a two-sided Wald test for each comparison performed by DESeq2[122]. Source data are provided as a Source Data file.

single neuropeptide gene, *ins-39* insulin/IGF1 hormone, that was consistently expressed higher in males during all developmental stages (Fig. 3b, c). We validated the higher expression of *ins-39* in males using a transcriptional *ins-39p::gfp* reporter array. Remarkably, adult males exhibited 32-fold higher mean fluorescence intensity than hermaphrodites (Fig. 3d–f), and L1 hermaphrodites showed essentially no fluorescence at all (Fig. 3d, e). The male-specific expression of *ins-39p::gfp* enabled us to distinguish L1 males from hermaphrodites both manually or automatically using a complex object sorter (COPAS)[88] (Fig. 3g, h), thus providing a needed early-stage male-specific marker as a powerful tool for high throughput isolation of males.

## INS-39 is highly enriched in select sensory neurons in males

INS-39 has recently been reported to be expressed in AFD neurons[89], but its male pattern suggests a broader expression. To determine the complete expression pattern for INS-39, we generated a nuclear-localized CRISPR/Cas9-engineered reporter allele [*ins-39*::SL2::GFP::H2B] (Fig. 4a). This reporter revealed a striking dimorphism across all stages: while hermaphrodite expression is low and restricted, males exhibit higher and broader neuronal GFP expression throughout development (Fig. 4b, c). We used "Neuro-PAL" (Neuronal Polychromatic Atlas of Landmarks) to identify cell types using color barcodes[90,91]. We found that in young adult

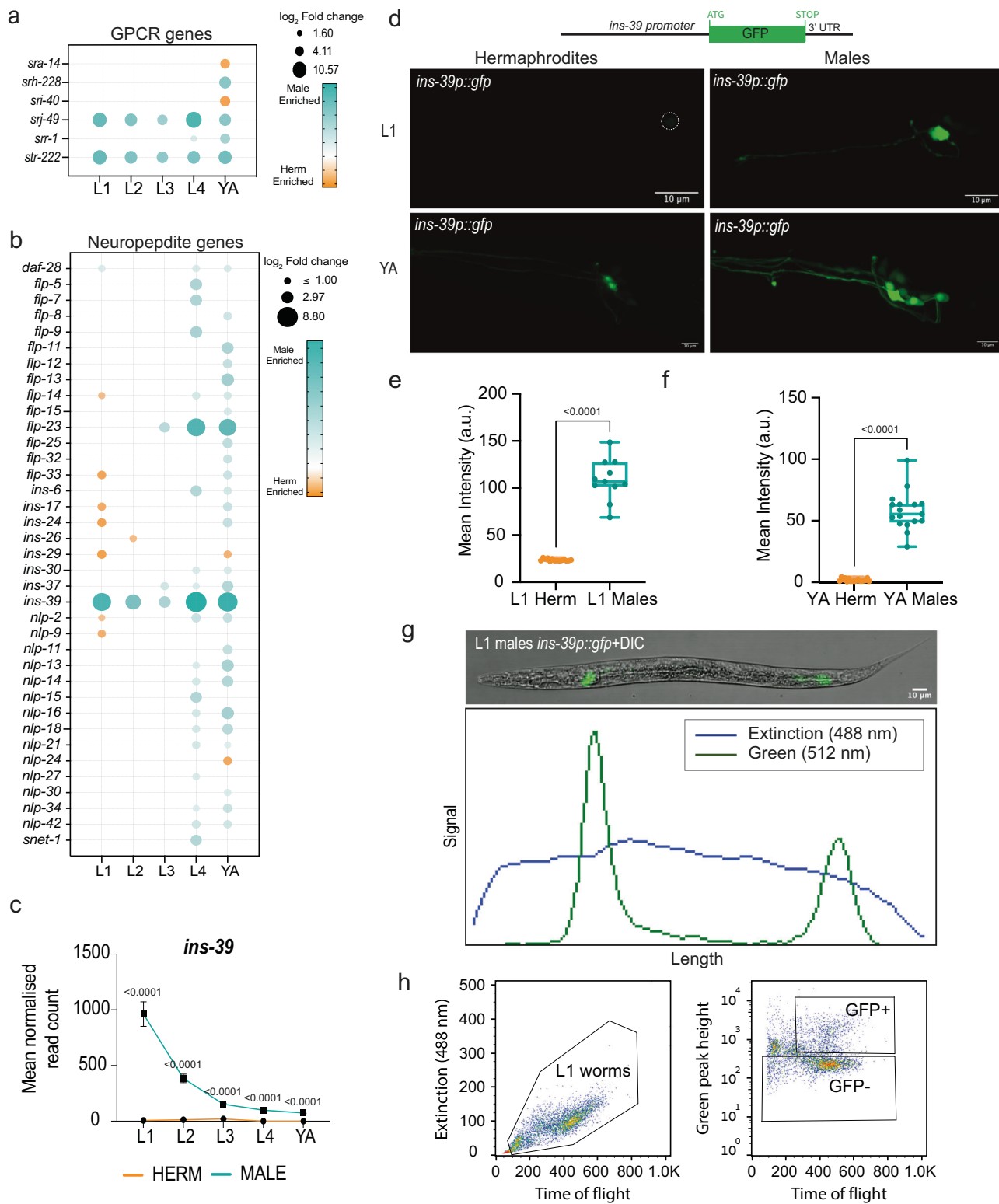

hermaphrodites, INS-39 was specifically expressed in two sets of neurons, AFD and ASK. In contrast, in males, INS-39 was expressed in five sets of neurons, AFD, ASK, ASJ, ASE, and AWC, all of which are ciliated sensory neurons with thermosensory, chemosensory, olfactory, electrosensory, and photosensory functions (Fig. 4d). Using fluorescent dye staining of head ciliated sensory cells and an AFD marker, we validated the higher GFP intensity in ASK, ASJ, and AFD neurons in adult males compared to hermaphrodites (Fig. 4e–g, Supplementary Fig. 7a, b). Taken together, the robust expression in

sensory neurons in males from the earliest stages of development suggests INS-39 is under tight and sex-specific regulatory control.

## Sex-related TFs instruct dimorphic INS-39 expression

Does the genetic sexual identity of the animal determine *ins-39* dimorphic expression? In *C. elegans*, sexual differentiation that gives rise to dimorphic features is controlled autonomously in every cell by the activity of the TRA-1 master regulator[92]. Previous studies have demonstrated that TRA-1 is either inactive or minimally expressed in

**Fig. 3 | Neuropeptide superfamily genes with sexually dimorphic regulation.** Bubble plot representation of GPCRs (**a**) and neuropeptide genes (**b**) differentially expressed in any of the five developmental stages of the two sexes. Bubble size represents $\log_2$ of fold change in expression of that gene (only genes that passed the filter padj <= 0.05, $|\log_2$ fold change$|>= 1$ and basemean >= 5 are plotted), and bubble color represents enrichment in either sex (male enrichment in cyan, hermaphrodite enrichment in orange). **c** Normalized expression values of *ins-39* across all developmental stages in both sexes (cyan: males, orange: hermaphrodites). $n = 4$ biological repeats per sample. Data are presented as mean values +/− SEM. **d** Schematic of the *ins-39p::gfp* expression reporter used in this study. Representative confocal micrographs of the expression pattern of the *ins-39p::gfp* reporter in head neurons at L1 and YA stage in both sexes (bottom). Scale bars represent 10 μm. Quantification of *ins-39p::gfp* fluorescence intensity from **d** in head neurons in both sexes at L1 (**e**) and YA stage (**f**). a.u, arbitrary units. In the box-and-whiskers graph, the center line in the box denotes the median, while the box contains the 25th to 75th percentiles of the dataset, whiskers define the minimum and maximum value with dots showing all points. $n = 11$ animals for L1 males and $n = 14$ animals for L1 herm **e** and $n = 17$ for **f** per group. **g** *ins-39p::gfp* enables male isolation using flow cytometry. Top, representative confocal micrographs of a whole L1 male expressing *ins-39p::gfp*. Bottom, extinction, and fluorescence profiles from *ins-39p::gfp* L1 males on the BioSorter profiler. The worm is oriented with its head towards the left side of the frame. The BioSorter profiler accurately captures the GFP intensity profile from the *ins-39p::gfp* expressing neurons (green channel), providing an effective means to sort L1 males from L1 hermaphrodites. Scale bars represent 10 μm. **h** Gating profiles for *ins-39p:gfp* expressing worms based on time of flight in the flow cytometer flow cell and extinction at 488 nm. The gate region was selected for L1 worms (time of flight vs. extension, left panel), and the gate region was selected for GFP- and GFP+ (time of flight vs. GFP intensity peak height, right panel). We performed a two-sided Mann-Whitney test in **e**, **f**. In **c**, adjusted p-values were calculated by a two-sided Wald test for each comparison performed by DESeq2[122]. Source data are provided as a Source Data file.

males, whereas it is abundantly expressed in hermaphrodites[61]. We first analyzed the *ins-39* locus for potential binding sites for TRA-1 and other TFs. We indeed identified a TRA-1 binding site in the *ins-39* first exon (Supplementary Fig. 7c) along with many other TFs listed in Supplementary Data 7 (see methods). To investigate whether INS-39 dimorphic expression is determined by TRA-1 we manipulated its expression by masculinizing or feminizing the entire nervous system and scoring the number of neurons that express INS-39. Animals with a sex-reversed nervous system were generated by pan-neuronal expression of the *fem-3* gene (masculinization)[93] or *tra-2(IC)* transgene (feminization)[94]. We found that masculinization of the nervous system significantly increased the number of INS-39-expressing neurons in hermaphrodites (Fig. 5a, b). Conversely, pan-neuronal feminization significantly decreased the number of INS-39-expressing neurons, specifically AWC, and to a lesser extent in ASE neurons (Fig. 5f, g, Supplementary Fig. 7d–f) but was insufficient to reduce INS-39 levels to that of wild-type hermaphrodites (AFD and ASK). These results suggest that neuronal TRA-1 functions to restrict INS-39 expression in hermaphrodites, but either additional factors or TRA-1 activity in additional tissues are required for the full scope of male expression. We first investigated the possibility that the DMD *mab-3* plays additional sex-specific roles that are not yet known, given the surprising expression dynamics our dataset revealed for *mab-3*, which included a robust peak at sexual maturation in both sexes but a greater level in males (Fig. 2d). We analyzed young adult *mab-3* hermaphrodite mutants and found that the number of INS-39::GFP-expressing neurons was significantly reduced (Fig. 5b, c). Interestingly, *mab-3* loss of function did not affect L1 hermaphrodite INS-39 expression in AFD neurons (Fig. 5c, d). As neuronal *ins-39* expression precedes that of *tra-1*[95], this result suggests an unknown, *tra-1/mab-3* independent regulatory mechanism that sets up the early sex-specific gene expression (Fig. 5e). In males, *mab-3* mutations significantly reduced the number of INS-39::GFP-expressing neurons from early development (Fig. 5g–i). Epistasis analysis of sex-reversed animals and *mab-3* mutants shows that *mab-3* loss of function suppresses the sex-reversal phenotype in hermaphrodites and enhances it in males (Figs. 5b, g), indicating that *mab-3* functions downstream of TRA-1 in regulating INS-39. This suppression was dramatic, but not complete (Fig. 5b, g). Taken together, the broader expression of INS-39 in masculinized hermaphrodites suggests INS-39 expression is actively repressed in many neurons by TRA-1, and that MAB-3 functions downstream to control INS-39 expression, along with additional factors. Our data set identified additional DMDs with interesting developmental expression (Fig. 2e). Screening previously published single-cell/bulk-sorted neuronal transcriptomic data sets for DMD genes expression that coincide with *ins-39* expression pattern[96–98], we found that *dmd-9* is highly expressed in hermaphrodites in *ins-39* expressing neurons, primarily in AFD (Supplementary Fig. 8a). Consistent with this, *dmd-9* loss of function

mutation was sufficient to suppress INS-39 GFP expression in AFD neurons in adult hermaphrodites, but not males, where we observed a global decrease that couldn't be attributed to a single neuron (Supplementary Fig. 8b, c). *dmd-9* loss of function did not affect L1 hermaphrodite INS-39 expression in AFD neurons (Supplementary Fig. 8d, e). *dmd-8* mutant animals did not show any significant change from wild-type animals, in line with its weak expression in INS-39-expressing neurons (Supplementary Fig. 8b). Thus, *dmd-9* is necessary for the expression of INS-39 in AFD neurons exclusively in adult hermaphrodites, whereas other TFs may be involved in early larval hermaphrodites and adult males. These results portray a complex and sexually dimorphic regulatory pathway for INS-39 expression. In hermaphrodites, TRA-1 suppresses INS-39 expression broadly in the nervous system, while *mab-3* and *dmd-9* act to maintain its expression in select neurons (e.g. AFD for *dmd-9*). In early larvae an unknown, *tra-1*, *mab-3*, and *dmd-9* independent regulatory mechanism exists (Fig. 5e). In males, in the absence of TRA-1, *mab-3* and/or additional factors function to promote the extremely high INS-39 expression in many sensory neurons (Fig. 5j). The emerging picture is of tighter regulation of INS-39 in hermaphrodites, whereas in males, a more complex regulatory network coordinates high INS-39 neuronal expression.

### Sex-specific roles of INS-39 in survival and stress response

INS-39 is a member of an expanded class of insulin-like peptides in *C. elegans* and functions through an evolutionary conserved insulin-like growth factor signaling pathway[99]. All 40 insulin-like peptides identified in *C. elegans* are thought to act through a single receptor, DAF-2, the homolog of the human Insulin-like growth factor receptor IGF1R. Insulin/IGF1 signaling (IIS) regulates dauer entry, behavior, aging, development, and fat accumulation[100]. Recently, it was discovered that high temperature represses INS-39 expression in the AFD neuron of hermaphrodites[89]. We, therefore, examined the changes in INS-39::GFP expression in the AFD neurons of both sexes when subjected to temperature shifts (25 °C and 15 °C) from the cultivation temperature (22 °C). As expected, we observed a significant reduction in INS-39 levels in AFD in hermaphrodites upon temperature shifts (Fig. 6a, b). However, INS-39 levels in AFD in males did not change with temperature shifts and remained high (Fig. 6a–c). We next tested INS-39 involvement in thermotaxis but found no apparent behavioral role for INS-39 in both sexes (Supplementary Fig. 9a, b).

*ins-39* was previously shown to play a role in nematode survival against peroxide stress through the insulin signaling pathway[89]. We confirmed the previously reported enhanced peroxide survival in *ins-39* mutant hermaphrodites grown and assayed at 20 °C or 25 °C (Fig. 6d, Supplementary Fig. 9c–f). However, unlike hermaphrodites, our results show that *ins-39* does not play a significant role in males, regardless of pre-exposure to higher (25 °C) or lower (20 °C) temperatures (Fig. 6d, Supplementary Fig. 9c–f). Furthermore, we

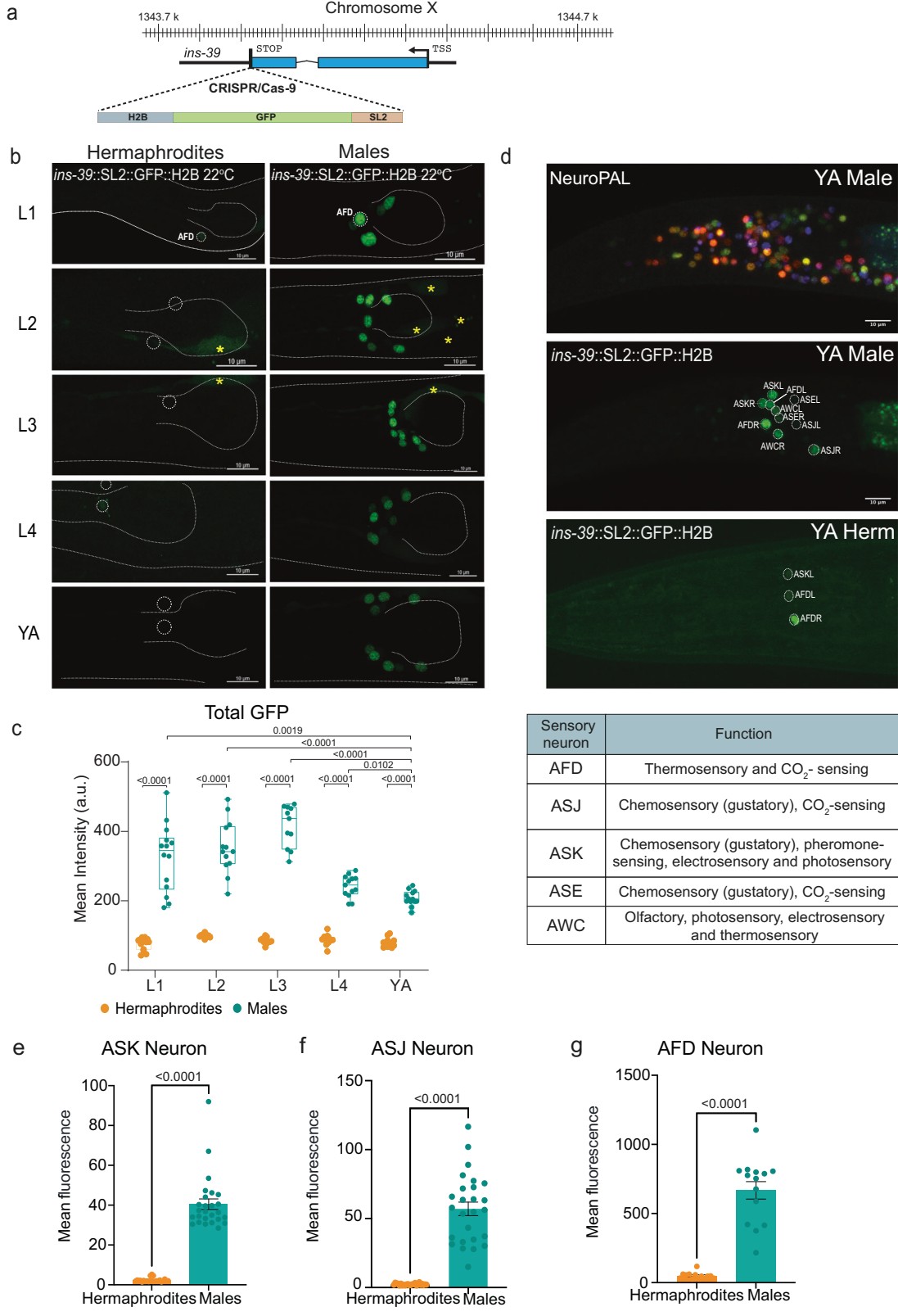

| Sensory neuron | Function |
|---|---|
| AFD | Thermosensory and $CO_2$- sensing |
| ASJ | Chemosensory (gustatory), $CO_2$-sensing |
| ASK | Chemosensory (gustatory), pheromone-sensing, electrosensory and photosensory |
| ASE | Chemosensory (gustatory), $CO_2$-sensing |
| AWC | Olfactory, photosensory, electrosensory and thermosensory |

observed that wild-type males exhibited lower survival rates against peroxide compared to hermaphrodites, although this phenotype was *ins-39* independent in males (Fig. 6d). Together, these behavioral assays suggest that INS-39 functions as a temperature and peroxide-stress sensor in hermaphrodites, but not in males.

Since *ins-39* levels are much higher in males already early during development, we reasoned this might provide some advantage over

hermaphrodites in coping with early life stress. Therefore, we focused on early life stress using two strategies. We first tested the ability of the animals to enter the dauer stage and found that wild-type males were more efficient in dauer transition than hermaphrodites, consistent with previous findings[101] (Fig. 6e). Unexpectedly, hermaphrodites mutant for *ins-39* became as efficient as wild-type males in dauer entry, while *ins-39* mutant males were similar to wild-type males (Fig. 6e),

**Fig. 4 | Sexually dimorphic expression pattern of INS-39. a** Schematic of the CRISPR/Cas9-edited *ins-39*(*syb4915*[*ins-39*::SL2::GFP::H2B]) reporter. **b** Representative confocal micrographs of the expression pattern of *ins-39*(*syb4915*) reporter in *C. elegans* across all developmental stages in both sexes. The head and pharynx are outlined in a white dashed line. Co-injection marker is indicated by a yellow asterisk. Scale bars represent 10 μm. **c** Quantification and comparison of mean total head GFP intensity of the *ins-39*(*syb4915*) reporter expression. L1 hermaphrodites: *n* = 13, L1 males: *n* = 14, L2 hermaphrodites: *n* = 12, L2 males: *n* = 13, L3 hermaphrodites: *n* = 12, L3 males: *n* = 11, L4 hermaphrodites: *n* = 11, L4 males: *n* = 13, YA hermaphrodites: *n* = 11, YA males: *n* = 13. In the box-and-whiskers graph, the center line in the box denotes the median, while the box contains the 25th to 75th percentiles of the dataset, whiskers define the minimum and maximum value with dots showing all points. a.u, arbitrary units. **d** Representative confocal micrographs showing genetically encoded multi-colored neuronal nuclei of a NeuroPAL worm used to identify the neurons expressing the *ins-39*(*syb4915*) reporter in young adult males and hermaphrodites. The table below lists the identified sensory neurons' established functions (www.wormatlas.org). The images in **b** and **d** were acquired using different confocal settings to capture the entire expression pattern. Scale bars represent 10 μm. Quantification of mean GFP intensity in ASK (**e**) hermaphrodites: *n* = 12, males: *n* = 13, ASJ (**f**) hermaphrodites: *n* = 12, males: *n* = 13, and AFD (**g**) hermaphrodites: *n* = 13, males: *n* = 14, neurons in both sexes using the *ins-39*(*syb4915*) reporter. In **e**–**g** data are presented as mean values +/− SEM. In **c**, **e**–**g**, we performed a two-sided Mann-Whitney test. Source data are provided as a Source Data file.

suggesting that *ins-39* expression has a dimorphic role in the dauer pathway.

Lastly, we tested L1 survivability. The IIS pathway has been shown to regulate L1 arrest, in which worms can cease growth and development as young larvae in the absence of food[102]. The survival of L1-arrested animals can be shortened or lengthened depending on the degree of IIS activity. With this in mind, we evaluated if *ins-39* had any sex-specific regulation of L1 survival. We first found that wild-type male worms exhibited a significantly lower survival rate (21.42% shorter) as arrested L1 larvae than hermaphrodites (Supplementary Fig. 9g). Interestingly, *ins-39* KO L1 animals of both sexes survived significantly longer than their wild-type controls, and males even more so than hermaphrodites (Fig. 6f). These results suggest a negative role for *ins-39* in the regulation of L1 survival, with a more critical role in males than hermaphrodites. Taken together, while in males *ins-39* levels seem mostly unresponsive to the environmental conditions tested in this study, in hermaphrodites *ins-39* modulation plays a role in context-specific response patterns.

## Discussion

Our transcriptomic atlas provides a much-needed molecular roadmap with which important developmental and molecular questions can be answered in a dimorphic context. We report a comprehensive gene expression atlas for both sexes of *C. elegans* throughout development which has been lacking until now. Our findings revealed 519, 63, 614, 2141, and 3564 genes that were differentially/dimorphically expressed in L1, L2, L3, L4, and YA, respectively, providing more comprehensive sex and stage datasets compared to previous findings[26,27,45,103,104]. Among previously published studies, only Kim et al. focused on the male-specific and nervous-system-specific expression of genes after sexual maturation. Our study takes a significant stride forward by focusing on a crucial aspect: the stage-specific transcriptomics of both sexes and a cross-analysis between them. We observed an increase (~12 times more) in the amount of identified dimorphic genes compared to previous work[27] which could be attributed to restrictions of using pseudo males to compare between stages in both sexes, resulting in a lower readout. It's important to note that our sequencing method, MARS-seq, has lower sensitivity and may not be as effective at capturing and detecting lowly expressed transcripts, which could account for the lower number of genes detected in this study[105]. Its advantages are higher true positive rates (accuracy in detecting DE genes) and price, compared to other RNAseq library preparation protocols.

Across different organisms, the expression levels of most genes change during development. In addition, sex bias in gene expression can vary greatly between and within species and depends on factors like species, tissue type, RNA-seq library preparation, sample size, and statistical criteria[12,16,106]. Several observations can be made when sex-enriched mRNAs at various stages are compared. It appears that sexual differentiation is first manifested by a small subset of sex-enriched genes in early larvae and a larger, more specific subset of mRNAs in later larvae throughout all developmental stages. Gene ontology enrichment of TFs during the pre-sexual maturation period showed a substantial enrichment for processes essential for promoting subsequent dimorphic development, as well as sexually dimorphic morphogenic processes. These TFs are likely to contribute to or drive the development of sex-specific traits and structures, which emerge later during sexual maturation. Additionally, there are more male-biased DEGs than hermaphrodite-biased DEGs. This male bias in differential gene expression has been observed in other organisms as well, ranging from plants to insects[16,107,108]. In humans, sex-biased gene expression is largely tissue-specific, and sex-biased genes exhibit nonrandom and tissue-specific genomic distribution[10].

Although the development of males and hermaphrodites is similar before sexual maturation, several crucial male cell fates are already determined at hatching. For example, three significant groups of male-specific blast cells give rise to the male mating structures[49]. Our list of male-enriched genes could point to additional genes that are involved in the regulation of male identity. The almost complete male-biased expression of neuronal genes could be explained by the size of the male nervous system, which contains 30 percent more neurons, as well as a larger connectivity network of chemical and electrical synapses. The unique neuro-peptidergic blueprint of males is an important outcome of this study. We find that neuropeptide coding genes are differentially expressed between the sexes and across development, with a strong bias towards higher male expression.

In rodents, Neuropeptide Y (NPY) expression in many brain areas under basal, unstressed conditions is lower in females than in males[109,110]. We show similar results for the insulin peptide *ins-39*, which exhibits strikingly higher and broader expression levels in males versus hermaphrodites. For the NPY system, it has been speculated that the lower expression levels might put females at a disadvantage in dealing with stress[110]. Our dauer entry and peroxide resistance assays show that for hermaphrodites the absence of *ins-39* improves their survivability, while it has no effect on males. The tight regulation of *ins-39* expression levels in hermaphrodites suggests that maintaining low levels is critical for the animal. Why males require high *ins-39* levels is unknown but may represent an evolutionary cost that comes as a tradeoff for an unknown advantage that awaits further research.

Interestingly, we found that the highest number of common DEGs was shared between L1 and YA. This perhaps points to these stages as critical windows of sexually dimorphic development, providing the needed flexibility to respond to environmental cues, a phenomenon observed in memory imprinting and early-life experience studies[111].

Very few DEGs in our dataset were common to all development stages. Out of the 769 TFs described so far in *C. elegans*[55], we identified a total of 218 TFs with temporal and sexual specificity throughout development. *ces-2*, for example, was consistently higher in males across all developmental stages. *ces-2* controls asymmetric cell division and mitotic spindle orientation in the neurosecretory motor (NSM) neurons lineage and is necessary to initiate programmed cell death[112,113]. Our study predicts that *ces-2* could be involved in establishing a male cell identity from the very first larval stage and, therefore crucial for further research.

Although our approach presents the most comprehensive gene expression dataset of dimorphic development to date, this

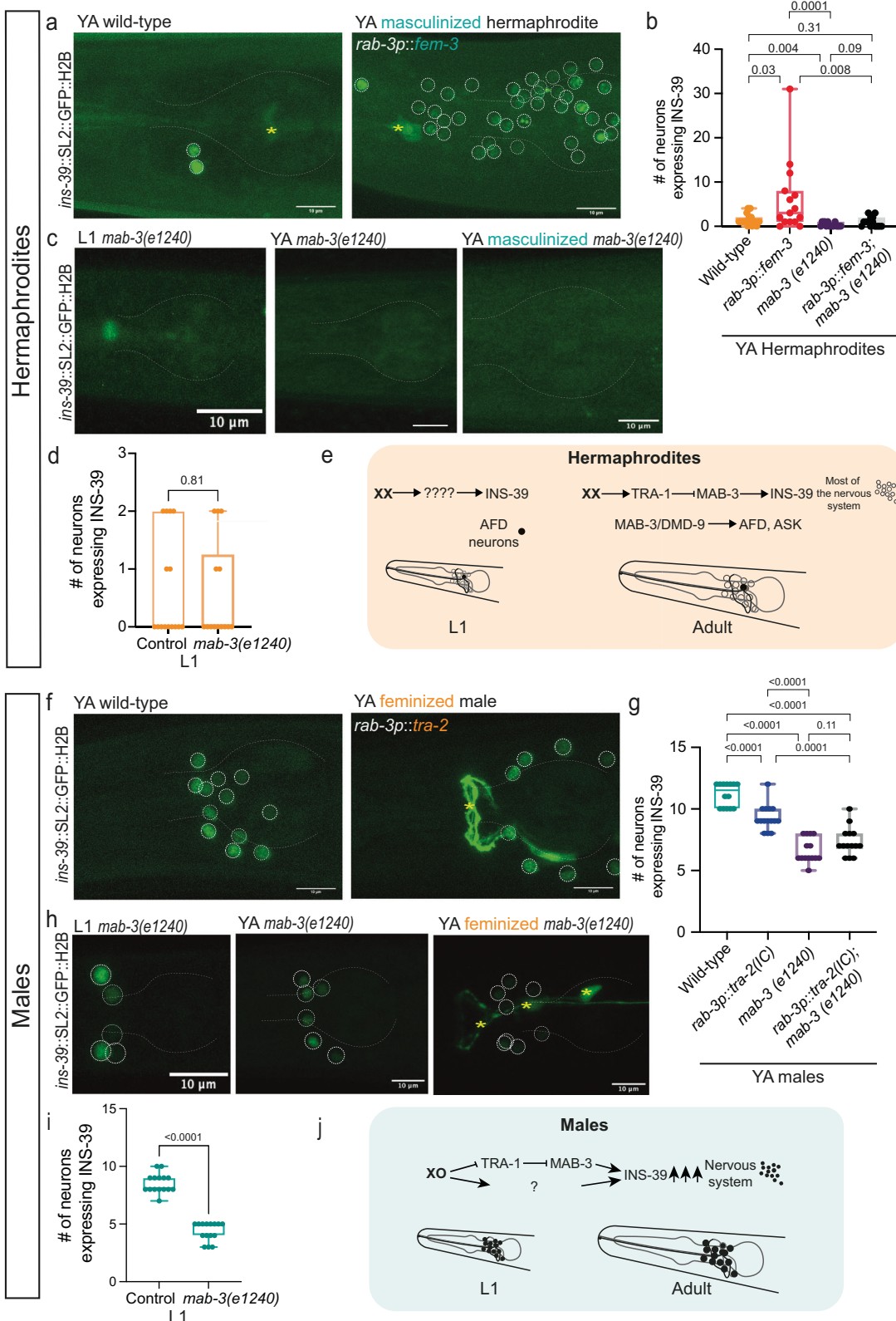

study is not full, for it was not designed to identify transcripts in embryonic stages[104], post-transcriptional regulation of genes, or dimorphic effects elicited by hormones as in mammals[114]. Nevertheless, this work does not only enrich the *C. elegans* community with additional methodologies and expression data but also offers 1047 new conserved dimorphic candidate genes with their associated human genetic disorders and traits that may underlie

the sexually dimorphic molecular characteristics of various human diseases.

## Methods

### *C. elegans* strains and maintenance

All *C. elegans* strains were cultivated as per standard methods[28]. Wild-type strains were Bristol, N2. *him-5(e1490)* or *him-8(e1489)* were treated

**Fig. 5 | TRA-1 and MAB-3 control INS-39 dimorphic neuronal expression.**
**a** Representative confocal micrographs of the *ins-39(syb4915)* reporter expression in a wild-type YA hermaphrodite and pan-neuronally masculinized YA hermaphrodites, expressing *rab-3p::fem-3*. Co-injection marker is indicated by a yellow asterisk. Scale bars represent 10 μm. **b** Quantification of the number of *ins-39(syb4915)* GFP-expressing neurons from wild-type, pan-neuronally masculinized, *mab-3 (e1240)* and pan-neuronally masculinized together with *mab-3 (e1240)* YA hermaphrodite. Wild-type: *n* = 15 worms, *rab-3p::fem-3*: *n* = 15 worms, *mab-3 (e1240)*: *n* = 15 worms, *rab-3p::fem-3; mab-3 (e1240)*: *n* = 14 worms. **c** Representative confocal micrographs of the *ins-39(syb4915)* reporter expression in a L1 hermaphrodite *mab-3(e1240)*, YA hermaphrodite *mab-3(e1240)* and pan-neuronally masculinized together with *mab-3 (e1240)* YA hermaphrodite. Scale bars represent 10 μm. **d** Quantification of the number of *ins-39(syb4915)* GFP-expressing neurons from wild-type L1 hermaphrodite: *n* = 15 worms and *mab-3(e1240)* L1 hermaphrodite: *n* = 14 worms. **e** A model depicting the molecular elements mediating INS-39 expression in hermaphrodite at L1 and YA stage. **f** Representative confocal micrographs of the *ins-39(syb4915)* reporter expression in a wild-type male and pan-

neuronally feminized male, expressing *rab-3p::tra-2(IC)*. Co-injection marker is indicated by a yellow asterisk. Scale bars represent 10 μm. **g** Quantification of the number of *ins-39(syb4915)* GFP-expressing neurons in wild-type: *n* = 14 worms, pan-neuronally feminized: *n* = 15 worms, *mab-3 (e1240)*: *n* = 14 worms and pan-neuronally feminized together with *mab-3 (e1240)*: *n* = 14 worms YA male. **h** Representative confocal micrographs of the *ins-39(syb4915)* reporter expression in L1 male *mab-3(e1240)*, YA male *mab-3(e1240)* and pan-neuronally feminized together with *mab-3 (e1240)* YA male. Scale bars represent 10 μm. **i** Quantification of the number of *ins-39(syb4915)* GFP-expressing neurons from wild-type L1 male and *mab-3 (e1240)* L1 male. *n* = 15 worms per group. **j** A model depicting the molecular elements mediating INS-39 expression in males at L1 and YA stages. In (**b**, **d**, **g**, **i**) the box-and-whiskers graph, the center line in the box denotes the median, while the box contains the 25th to 75th percentiles of the dataset, whiskers define the minimum and maximum value with dots showing all points. We performed a two-sided Mann-Whitney test for each comparison. Source data are provided as a Source Data file.

as controls for strains with these alleles in their background. Worms were grown on nematode growth media (NGM) plates seeded with *E. coli* OP50 bacteria as a food source. The sex and age of the animals used in each experiment are noted in the associated figures and legends. All transgenic strains used in this study are listed in Supplementary Data 8.

### Generation of *C. elegans* male cultures
To generate pure male populations, temperature-sensitive mutation (*y1*) in the dosage compensation gene, *dpy-28* along with *him-8(e1489)* mutations were employed[36]. *dpy-28(y1) him-8(e1489)* worms were cultured at 15 °C in 15 cm NGM plates seeded with OP50. Embryos were isolated by hypochlorite treatment of gravid adults collected from 3-4 NGM plates. The obtained embryos were left to hatch for 14-16 hours on a foodless NGM plate at a restrictive temperature of 25 °C. At the restrictive temperature, most hermaphrodites die as embryos or L1 larvae, whereas the XO males are unaffected. After 14-16 hours, L1 larvae and unhatched embryos were collected in 2 ml M9 buffer and pelleted at 3000 rpm for 1 min. Approx. 5000 L1/embryos were placed onto one side of the plate (area without food), roughly 1.5 cm from the food on an NGM plate (Supplementary Fig. 1). Depending upon the density of the pellet, the number of plates was increased accordingly. Plates were incubated at RT for 1-2 hours allowing the L1 animal to crawl towards food. Following incubation, the area where L1/embryos were placed was removed from the plate. L1 worms that were on the food were collected using 1 ml M9 buffer and washed 5 times with M9 buffer to get rid of bacteria. The L1 animals were counted and about 25,000 worms were placed on 15 cm NGM plates with OP50 at 25 °C until they reached the desired time point. The percentage of males obtained was counted for each set of experiments. A schematic representation of this protocol is illustrated in Supplementary Fig. 1.

### Automated worm tracking
Day 1 adult *him-8(e1489)* males and *dpy-28(y1) him-8(e1489)* males were assayed for their speed/locomotion and body area. Adult males from both groups were placed onto an NGM plate seeded with 30 μL of OP50 bacteria. 2-4 worms were placed onto food and a 1.5 cm diameter plastic ring was placed around them to prevent crawling away from the camera's field of view. The plate was placed inside the tracker and worms were left to habituate for 10 min. We used WormLab automated tracking system (MBF Bio-science)[115], to track and record the worms for 2 min at room temperature. To recover the worm's contour and skeleton for phenotypic analysis, the collected videos were segmented, and body area, speed, track length, mean amplitude, cumulative reversal time, and cumulative forward time metrics were exported. Prism 9 (GraphPad) version (9.5.0) was used for statistical analysis.

### Body length and width measurement
Day 1 adult *him-8(e1489)* males and *dpy-28(y1) him-8(e1489)* males were imaged in a brightfield (DIC) channel using a Zeiss LSM 880 confocal microscope with a 20x objective lens. The images were imported to ImageJ/Fiji software, version 2.3.0/1.53q, and a line was drawn from the tip to the tail, or across the width of each worm. The length of the line was measured and statistically analyzed.

### Mating behavior assay
Mating assays were conducted based on previously established methods[21,116]. 15-20 early L4 males from the test group and 15–20 early L4 *unc-31(e928)* hermaphrodites were separated from a mixed-stage population and were transferred to separate fresh OP50 NGM plates and incubated at 20 °C until they reached sexual maturation. On a fresh plate seeded with a thin lawn of OP50 bacteria, 10 adult *unc-31(e928)* hermaphrodites were transferred. *unc-31(e928)* hermaphrodites do not move a lot, allowing for an easy capturing of male behavior. A single virgin male from the test group was added to this lawn. Animals were kept under observation, and the course of events was recorded using a Zeiss Axiocam ERc 5 s mounted on a Zeiss stemi 508 for a period of 15 minutes, or until the male ejaculated, whichever occurred first. Mated males and hermaphrodites were discarded from the plate. A new hermaphrodite was then added to the lawn to keep the number of hermaphrodites similar between experiments. The video was analyzed, and the males were scored for their time until successful mating, contact response, and vulva location efficiency. Contact responses were defined as mail tail apposition and the start of backward movement on the hermaphrodite's body. Percentage response to contact was calculated using the formula (the number of times a male showed contact response/the number of times the male contacts a hermaphrodite via the rays * 100[21,116]. Vulva location efficiency (L.E.)[117], was calculated using the formula 1/number of major passes or hesitations at the vulva until ejaculation or 15 min time window.

### RNA Isolation, and library preparation
More than 2000 worms from each developmental stage and sex were collected from agar plates using an identical isolation protocol and were washed 5 times using M9 buffer to get rid of bacteria. Each set contained 4 biological replicates. Worms pellet were collected in 500 μL of Trizol and flash frozen in liquid nitrogen and stored at −80 °C until further processing. Total RNA was prepared from the frozen worm pellets following the instructions for the TRIzol LS (Invitrogen) protocol. After the isopropanol precipitation step, the RNA was resuspended in the extraction buffer of the RNA isolation kit (PicoPure, Arcturus), and further isolation was carried out following the manufacturer's instructions. When starting with a smaller number of worms, this two-step purification technique aids in getting RNA of good

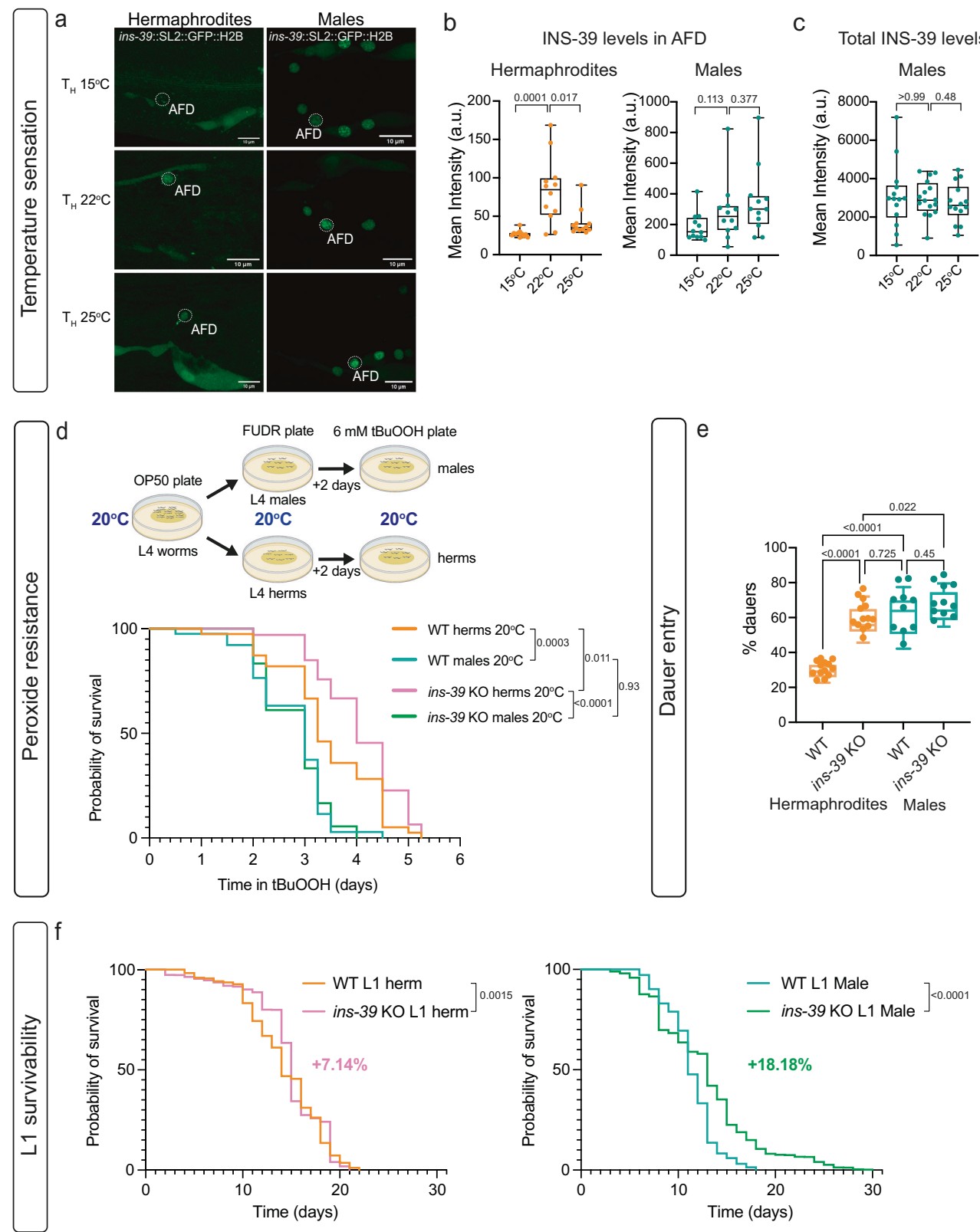

quality. RNA concentration was determined by Qubit Fluorometer (Invitrogen, Model #4), and RNA integrity was checked using a Bioanalyzer (Agilent). Only samples showing RIN[e] ≥ 8 were further processed. To create RNA-seq libraries for expression profiling, a bulk version of the MARS-Seq procedure was employed[39,40]. Briefly, reverse transcription was used to barcode and pool 18 ng of input RNA from each sample. The pooled samples underwent second-strand synthesis

after Agencourt Ampure XP beads cleanup (Beckman Coulter), and they were linearly amplified by T7 in vitro transcription. The resulting RNA was fragmented and then converted into a final library by tagging the samples with Illumina sequences during ligation, RT, and PCR. Finally, libraries were quantified by Qubit Fluorometer Model #4 and 2200 TapeStation nucleic acid system followed by qPCR for *lmn-1* gene[39,40]. Sequencing was carried out using an Illumina Nextseq 75

**Fig. 6 | Functional characterization of the role of INS-39. a** Representative confocal micrographs of the *ins-39(syb4915)* reporter expression in worms grown at 22 °C and then transferred to 25 °C or 15 °C for 6 hours. Scale bars represent 10 μm. **b** Quantification of mean GFP intensity from **a** in AFD in hermaphrodites and males. hermaphrodites 15 °C: *n* = 11 worms, males 15 °C: *n* = 12 worms, hermaphrodites 22 °C: *n* = 12 worms, males 22 °C: *n* = 12 worms, hermaphrodites 25 °C: *n* = 12 worms, males 25 °C: *n* = 12 worms. a.u, arbitrary units. **c** Quantification of mean GFP intensity from (**a**) in *ins-39(syb4915)* GFP-expressing neurons in males. males 15 °C: *n* = 16 worms, males 22 °C: *n* = 14 worms, males 25 °C: *n* = 14 worms. a.u, arbitrary units. **d** Schematic of the survival assay on 6 mM tBuOOH and cumulative survival graph on 6 mM tBuOOH of wild-type and *ins-39* KO (*ety9*) hermaphrodites and males. *n* = 40 worms per group. To assess and compare survival rates, a two-sided Kaplan-Meier method was employed. Statistical significance was determined using the Log-rank (Mantel-Cox) test in Prism (GraphPad). FUDR, 5-fluoro-2′-deoxyuridine. tBuOOH, tert-butyl hydroperoxide. Created with Biorender.com.

**e** Quantification of dauers percent in wild-type and *ins-39* KO (*ety9*) strain in hermaphrodites and males habituated at 27 °C (see methods). Wild-type hermaphrodites: *n* = 13, *ins-39* KO (*ety9*) hermaphrodites: *n* = 13, Wild-type males: *n* = 10, *ins-39* KO (*ety9*) males: *n* = 11 biological repeats. **f** Cumulative L1 survival graph of wild-type and INS-39 KO (*ety9*) hermaphrodites and males. Graphs show the composite of three independent experiments. The n for each group in the final datasets was as follows: Wild-type L1 Herm *n* = 2676, INS-39 KO (*ety9*) L1 Herm *n* = 1421, Wild-type L1 Male *n* = 2241, INS-39 KO (*ety9*) L1 Male *n* = 5453. To assess and compare survival rates, a two-sided Kaplan-Meier method was employed. Statistical significance was determined using the Log-rank (Mantel-Cox) test in Prism (GraphPad). In (**b**, **c**, **e**) the box-and-whiskers graph, the center line in the box denotes the median, while the box contains the 25th to 75th percentiles of the dataset, whiskers define the minimum and maximum value with dots showing all points. We performed (in **b**, **c**, **e**) a two-sided Mann-Whitney test for each comparison. Source data are provided as a Source Data file.

cycles high output kit on NextSeq500 with paired-end sequencing. The median, mean and standard deviation of the sequencing depth of all samples were 14.08, 15.02, and 5.38 million reads.

### RNA-seq data analysis

Raw next-generation sequencing (NGS) data in the FASTQ format were analyzed using a User-friendly Transcriptome Analysis Pipeline (UTAP) (https://utap.wexac.weizmann.ac.il/)[118]. Briefly, reads were trimmed using "cutadapt"[119] and mapped to the *C. elegans* reference genome WBcel235 using STAR v2.4.2a[120]. UMI counting was done using HTSeq-count[121] after marking duplicates in union mode. Differential gene expression analysis and normalization of the counts were performed using DESeq2[122] using the rld (log$_2$ normalized) gene expression values on a per-developmental stage basis between sexes. Raw p-values were adjusted with the Benjamini and Hochberg method[123]. A threshold of p-adjusted ≤ 0.05, |log$_2$FoldChange| ≥ 1 and baseMean >= 5 was used to identify genes with significant differential expression. The WormBase gene set enrichment analysis tool was used for functional annotation of gene ontology terms that are over-represented in differentially expressed genes for each developmental stage and sex[53,87]. A q-value threshold of 0.1 was used for every analysis. Jvenn[48] was used to create diagrams.

### Gene conservation analysis between *C. elegans* and Human

Human orthologs for the *C. elegans* genes and their associated Online Mendelian Inheritance in Man (OMIM) human-disease phenotypes were downloaded from OrthoList 2 (OL2)[124], a compendium of *C. elegans* genes with likely human orthologs (http://ortholist.shaye-lab.org/). The genes from our RNA-seq datasets were overlapped with the list from OrthoList 2 and were incorporated into the analysis.

### Microscopy

Worms were anesthetized on a drop of 200 mM sodium azide (NaN3, in M9 buffer) mounted on a freshly prepared 5 % agarose pad on a glass slide. Images were acquired using a Zeiss LSM 880 confocal microscope with a 63x objective lens unless otherwise noted and processed using ImageJ/Fiji software version 2.3.0/1.53q[125]. All images of the same group were imaged under identical microscope settings. For the expression of reporters, representative images with maximum intensity projections are shown for all channels. Individual neurons were identified through reporter/stain expression, and the z-plane exhibiting the most robust signal was selected for measuring fluorescence intensity using ImageJ version 1.52p. Figures were prepared using Adobe Illustrator V 25.0.1. For imaging NeuroPAL worms, channels were pseudo-colored in accordance with ref. 91.

### BioSorter analysis

*C. elegans* strains *ins-39p::gfp* or *him-5* (control) were grown and maintained at 20 °C. L1 synchronized nematode cultures were obtained by standard bleaching protocols. Eggs were allowed to hatch

overnight on foodless (i.e., without a bacterial lawn) 60 mm NGM plates, then L1-arrested larvae were collected in 50 ml conical tubes and were resuspended in S-basal at a final density of about 2000 worms/ml. Worm sorting was performed on a BioSorter equipped with a 250 FOCA and S-basal as the sheath fluid. To isolate L1 males, the *ins-39p::gfp* strain was used (GFP positive in head neurons). Non-fluorescent *him-5* strain was used to set gates for GFP-positive worms and to exclude auto-fluorescent worms. Sorted worms were collected onto a 60 mm OP50 plate. Graphs were prepared using FlowJo V 10.8.1.

### Neuron identification using NeuroPAL

For imaging NeuroPAL worms, the protocol followed is in accordance with ref. 91. For INS-39 CRISPR GFP neuronal identity, colocalization with the NeuroPAL landmark strain OH15262 harboring the *otIs669* transgene was used to determine the identity of all neuronal expression.

### DiD staining

Age synchronized worms were washed twice with 500 μl M9 buffer in a 1.5 ml microfuge tube and incubated in the dark for 1 h in 1 ml M9 containing 5 μl DiD dye (Vybrant™ DiD Cell-Labeling Solution, ThermoFisher) at -20-30 rpm. The worms were then centrifuged, and worm pellet was transferred to a fresh plate and animals were let to move onto the bacterial lawn.

### RT-PCR

Total RNA was extracted from 10 to 20 μl of age-synchronized packed worms' pellet (washed three times with M9 buffer) from relevant genotypes/developmental stages using the protocol described above under "RNA Isolation, and library preparation". The isolated total RNA was then converted to cDNA using SuperScript™ IV first-strand synthesis system (Invitrogen, Catalog # 18091050) with random hexamers. The concentration of cDNA was determined using NanoDrop One/OneC microvolume spectrophotometers and fluorometer. qPCR reactions were set up in 10 μl volume using Fast SYBR™ Green Master Mix (Applied Biosystems, Catalog # 4385612) as per manufacturer protocol using StepOnePlus™ Real-Time PCR System. Primers used are detailed in Supplementary Data 8. Ct values were retrieved and relative quantification of the expression of the target genes was performed utilizing the $2 - \Delta\Delta Ct$ method. *pmp-3* or *tba-1* were used as housekeeping genes.

### CRISPR/Cas9-mediated genome editing

To generate the *ins-39(ety9)* deletion strain, the previously described CRISPR/Cas9 genome engineering protocol was employed[126]. Briefly, tracrRNA and three crRNAs targeting the *dpy-10* locus and the boundaries of the *ins-39* locus were combined with a recombinant cas9 (IDT) and supplemented with ssODN repair template to introduce a

dominant mutation into the *dpy-10* locus[126] and a second ssODN composed of the 5′ and 3′ flanking sequences of the *ins-39* locus boundaries (see Supplementary Data 8 for sequences). Rol/dpy F1 progeny were singled and screened by PCR using the primers ATAG CAGAACATGGGCATCC and AACCGTTGGGTATTTGACCA, which amplify a 735 bp product only in *ins-39* deleted animals. Plates with a correct PCR signal were homozygozed and Sanger-sequenced to validate the accuracy of the *ety9* edit. The resulting strain was backcrossed 2 times to lose the *dpy-10* mutation.

The *syb4915*, *syb4956*, and *syb4992* allele were generated by SunyBiotech.

### Promotor analysis
*ins-39* promotor region was extracted from WormBase by taking 1000 bp upstream and 500 bp downstream of the transcription start site which included the whole protein-coding part of the gene. *tra-1* binding sites were derived into the matrix from a published study[127] using the genomatix genome analyzer tool and *C. elegans* transcription factors binding sites were downloaded from CisBP (http://cisbp.ccbr.utoronto.ca/). The promotor region binding sites were determined by using the RSAT matrix-scan search tool.

### Thermotaxis assay
Age-synchronized animals were sex-separated at late L4-YA stages and moved to 15 °C, 22 °C, and 25 °C for 6 hours. The animals were then washed and placed onto the thermotaxis plate. A modified steep linear gradient was maintained on a 10 cm plate using an aluminum plate with a hot plate (set at 27 °C) on one side and an ice pack on the other[128]. Temperature on the agar surface was measured using a temperature gun. The gradient was allowed to stabilize for 5 minutes, after which the animals were introduced at 22 °C isotherm and allowed to roam freely for 20 minutes. The animals were killed by quickly transferring to −20 °C for 5 minutes. Animals were scored for positive or negative thermotaxis as 1 cm from the center of origin towards either side. The thermotaxis index was calculated for each experimental group as:

$$Thermotaxis\ Index = \frac{\#\ of\ animals\ on\ warmer\ side - \#\ of\ animals\ on\ colder\ side}{total\ \#\ of\ animals} \quad (1)$$

### Dauer formation assay
Dauer formation assay induced by high temperatures was conducted following previous protocols[101], with slight modifications. 50 gravid adults that had been continuously raised at 15 °C were placed on OP50 plates for 6 hours at room temperature. The hermaphrodites were removed, and the laid eggs were incubated at 27 °C for 44 hours, while control plates were kept at room temperature for the same duration. To examine the presence of dauers (active, thrashing animals) and non-dauers (inactive, non-thrashing animals), we flooded the assay plates with 2 mL of 1% SDS solution, swirled them, and incubated them for 15 minutes at room temperature. The number of dauers and non-dauers animals were recorded and analyzed. All experiments were conducted in 3 independent biological trials.

### Survival assays in tert-butyl hydroperoxide
The survival assay was conducted on solid agar plates containing tert-butyl hydroperoxide at a final concentration of 6 mM, following previous protocols[89], with slight modifications. To prepare the assay plates, tert-butyl hydroperoxide was added to molten agar maintained at 55 °C, and 12 ml of the mixture was poured onto 60 mm NGM agar plates. The plates were left to dry at room temperature for 2 days and then seeded with 200 µl of a 5x concentrated OP50 culture that had been grown overnight, pelleted, and resuspended in M9. The worms

were cultured on standard OP50 plates until they reached the L4 developmental stage and then separated by sex into groups of up to 100 on plates containing 10 µg/ml 5-fluoro-2′-deoxyuridine (FUDR). For each genotype and sex, 40 Day-2 adults were transferred to assay plates at the specified temperature. The worms were monitored every 6–12 hours and scored as alive, dead, or censored until all worms had died. A worm was considered "dead" if it did not show any visible movement in response to a gentle touch from a poking lash, while worms that were lost or had desiccated on the side of the plate were considered "censored." Manual scoring was conducted, and the resulting data was plotted and analyzed. All experiments were conducted in 3 independent biological trials. To compare the survival rates a two-sided Kaplan-Meier method was used. The statistical significance was calculated using the Log-rank (Mantel-Cox) test in Prism 9 (GraphPad) version (9.5.0).

### L1 survival assay
L1 survival assay was conducted as described previously with slight changes[102]. Briefly, L1 animals were obtained from well-fed *C. elegans* plates from the indicated genotype, employing the same method to isolate males as described above under "Generation of *C. elegans* male cultures". This was done to keep all the parameters similar across all the experimental groups. The L1 worms were collected using M9 from the foodless plate and were diluted to a final concentration of no more than 10 L1 worms/µl. This was done because the survivability of *C. elegans* at L1 arrest depends on the density of worms[102]. Starvation cultures were placed in 40 ml glass vials and incubated on a rocker-shaker at 20 °C at low speed. Each day, aliquots containing at least 50 L1 arrested worms from each group were removed and plated on a foodless plate. The live/dead and total number of worms were recorded each day. The experiments were combined for analysis and represented using survival graphs in Prism 9 (GraphPad) version (9.5.0). In the final datasets, the animal count was contingent upon the maximum number of days of surviving animals. The 'n' value for each group in the final datasets was as follows: Wild-type L1 Herm $n = 2676$, INS-39 KO L1 Herm $n = 1421$, Wild-type L1 Male $n = 2241$, INS-39 KO L1 Male $n = 5453$. To assess and compare survival rates, a two-sided Kaplan-Meier method was employed. Statistical significance was determined using the Log-rank (Mantel-Cox) test in Prism 9 (Graph-Pad) version (9.5.0).

### Quantification and statistical analysis
Information about quantification, statistical testing, and sample size can be found in each figure legend and relevant method section.

### Transgenic strains/molecular cloning
To generate pan-neuronal masculinized strains *fem-3* was expressed under *rab-3p*. See Supplementary Data 8 for primers.

### Statistics & reproducibility
No statistical method was used to predetermine the sample size. No data were excluded from the analyses. The experiments were not randomized. The Investigators were not blinded to allocation during experiments and outcome assessment.

### Reporting summary
Further information on research design is available in the Nature Portfolio Reporting Summary linked to this article.

## Data availability
The sequence data generated in this study have been deposited in the GEO database and are available under accession code GSE237937. All data generated or analyzed in this study are included in the manuscript along with supplementary information files and Source Data files. Source data are provided in this paper. The RNA-seq datasets

associated with this article are included in Supplementary Data 2. Source data are provided in this paper.

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

## Acknowledgements

We thank members of the Oren-Suissa lab for their critical insights regarding the manuscript. We thank Dr. Shifra Ben-Dor at Bioinformatics Unit, Life Sciences Core Facilities of Weizmann Institute of Science for helping with promotor analysis. RNA-seq library preparation was done with critical advice from Dr. Hadas Keren-Shaul, Dr. Merav Kedmi and Dr. David Pilzer from the Genomics Sandbox unit at the Life Science Core Facility of Weizmann Institute of Science. We are grateful to Patrick Laurent for sharing strains. We thank Ronen Hayun, Maayan Maron, Eli Hotoveli, Anastasia Zarankin, Avigail Gedanken, Yarden Tala, and Moshiko Hafzadi at Design & Development of Weizmann Institute of Science for the website development. Some strains used in this study were obtained from Caenorhabditis Genetics Center (CGC), which is funded by the NIH Office of Research Infrastructure Programs (P40 OD010440). We thank WormBase, an online biological database for *C. elegans*, which is supported by Grant U41 HG002223 from the National Human Genome Research Institute at the NIH, the UK Medical Research Council, and the UK Biotechnology and Biological Sciences Research Council. Figures 1a, 1c, 6d, Supplementary Fig. 1, and Supplementary Fig. 9c,e were created with BioRender.com. M.O.S. acknowledges financial support from the European Research Council ERC-2019-STG 850784, Israel Science Foundation grant 961/21, Dr. Barry Sherman Institute for Medicinal Chemistry, Sagol Weizmann-MIT Bridge Program, and the Azrieli Foundation. M.O.S. is the incumbent of the Jenna and Julia Birnbach Family Career Development Chair. R.H. is greatly thankful to the Council of Higher Education, Israel for the PBC postdoctoral fellowship.

## Author contributions

R.H. conducted and analyzed the experiments. H.S. contributed to the imaging, crossing, and analysis of INS-39 expression. E.L. helped in imaging NeuroPAL strains. Y.S. conducted CRISPR/Cas9-mediated genome editing. S.K. conducted thermotaxis assays. S.K. and Y.S. carried out the cold tolerance assay. G.S. and H.G. carried out the bioinformatics analysis, under the supervision of O.R. and M.O.S. M.O.S. supervised and designed the experiments. R.H., Y.S. and M.O.S. wrote the paper.

## Competing interests

The authors declare no competing interests.
