## [Peer Review File · Nature Communications]

REVIEWER COMMENTS

Reviewer #1 (Remarks to the Author):

This study introduces an RNAseq dataset of larval and young adult animals from all developmental stages in both *C. elegans* sexes. While other RNAseq datasets exist from subsets of these conditions, this represents the most comprehensive dataset in *C. elegans*. Additionally, provides solutions to the difficulty in isolating populations of *C. elegans* males, first to generate the RNAseq dataset, and then alternative methods based on what was learned from the dataset itself. This dataset enables the authors to identify dimorphically-expressed genes, such as the insulin-like peptide INS-39. The authors attempt to demonstrate and interpret some functional relevance of this dimorphically-expressed insulin, although the effects seem mild and difficult to interpret (see final comment below). This is the only real weak point in the presentation of a dataset that will surely be of broad interest for the field.

Detailed Comments:

p.13: The claim that srj-49 is a noncoding RNA is unsubstantiated, and an unlikely explanation based on the data actually provided. If srj-49 is spliced (even if it is not translated), the SL2::GFP::H2B should be expected to be expressed. If the actual mapped RNAseq reads suggest that srj-49 transcripts only appear unspliced (ie no evidence of exon/exon junctions at all), then maybe this hypothesis can stand, but the amplification of cDNA in S4B suggests otherwise.

Fig 4: Panel B seems to indicate little-to-no expression of INS-39 in the hermaphrodite (the only indicated expression seems to be faint AFD in L1, and then nothing perceptible in any other stage besides maybe something on top of the pharynx in L4). However, panel D shows clear expression in multiple YA hermaphrodite neurons. These two panels use the same endogenous reporter, so they should presumably show the same expression pattern.

Fig 5: With so few expressing neurons, the seemingly weak effect especially in panel H, and the fact that ins-39 was already combined with NeuroPal in Fig 4, it would be more satisfying to have neuron IDs rather than simply counting expressing cells in this figure. Also, entitling this figure “spatial localization” seems to imply that INS-39 is mis-localized by changing sexual identity rather than differentially regulated. A final minor point, the GFP reporter is labeled as INS-39::GFP in all images of this figure, where it should presumably be indicated as SL2::GFP::H2B.

Fig S6: Fluorescence images should be shown for the dmd-8 and dmd-9 mutants. This figure and the results are a bit hard to interpret because the expectation seems to be that dmd-9 is regulating INS-39 only in AFD (maybe AWC?) but the quantification in panels D and E doesn't seem to have been

performed in a neuron-specific manner (I was unable to find sufficiently specific details to confirm this in the Methods).

p.24: Figure 4 indicates that INS-39 is not expressed in hermaphrodite ASJ at any stage. Thus, the rationale for invoking ASJ's role in cold tolerance (and the hermaphrodite improved survival rate that is ins-39-dependent), is unclear.

Fig. 6G: The text indicates that survival is "significantly longer," but no statistics are shown or indicated in the figure legend or methods.

p. 25: The conclusions "we suggest that the lower expression levels of INS-39 in hermaphrodites is fine-tuned to critically respond to environmental changes, providing a survival advantage in the face of unfavorable conditions. In contrast, the higher expression in males might hinder its functionality, creating a ceiling effect," are a bit confusing. For instance, in the cold tolerance assay, ins-39 KO has a mild (positive) effect on hermaphrodite survival but none in males. Additionally, ins-39 KO hermaphrodite enter dauer at a higher rate in response to stress; this could be considered to be an advantage for survival. The claim that the lower expression levels critically respond to environmental changes would suggest that hermaphrodites should always be compromised by loss of ins-39? In addition, I would suggest avoiding claims about the functionality of variations in expression level (ie invoking a ceiling effect) when all experiments show all-or-none changes and with conflicting results such as these. These interpretations are also presented in the Discussion section.

Reviewer #2 (Remarks to the Author):

The authors describe the results of whole animal RNAseq transcriptome analysis for *C. elegans*. They have analyzed RNA samples prepared separately from each of the four developmental stages plus young adults, and separately for each sex (10 averaged samples in all). Transcriptome comparison of the sexes of *C. elegans* has been carried out before. What is new and valuable here is the authors have developed a method to separate males and hermaphrodites at early larval stages. Thus, comparison of the transcriptomes of the two sexes at early larval stages has not been done before. The transcriptome data of Supplementary Table 2 is valuable.

A difficulty with this paper is that there is insufficient comparison of these new results with previous data. The authors note a number of limitations of earlier work, but their comparison is inaccurate in one important respect. They state in the introduction and again in the discussion that earlier results were limited by small sample sizes, due to the necessity of, in some cases, picking males from a mixed

population by hand. The RNAseq method is so sensitive that small sample size is not a limitation. In fact, Kim et al., (2016), who picked males by hand and thus have a pure separation of the sexes, detected expression of more genes (21,143, including 17,967 protein coding) than the present work (14,185). This is particularly surprising since Kim et al did not include data from L1 or L2 larval stages.

Considering what is new here, transcriptomes of L1 and L2 larvae, comparison of the sexes at early larval stages should have been emphasized in the analysis. While the authors do describe their results, they highlight no particular significant findings. Rather, they focus on male-specific and nervous-system specific expression. Since this is precisely the question also analyzed by Kim et al (2016), the present authors need to justify this choice. A comparison of the gene lists should have been included. Was anything new found? A few spot checks in fact uncovers discrepancies—Kim et al, for example, do not show *ceh-13* or *nob-1* as male-enriched transcription factors. A spot check finds a discrepancy within this manuscript itself: *ttx-1* is described as expressed higher in males during sexual maturation, but unless this reviewer misunderstands the data, expression of *ttx-1* is in fact higher in L1 males (201 average normalized reads) than in L3 males (39 average normalized reads) or L4 males (43 average normalized reads).

For the second part of the paper, the authors select one gene enriched in male expression, *ins-39*, for further study. But they don't learn anything in particular about it and there is no justification for this direction of the study. They do not learn the mechanistic basis of its male-specific expression or the functional reason the gene is expressed in a few male-specific neurons.

Reviewer #3 (Remarks to the Author):

This is a very comprehensive analysis of sexually dimorphic gene expression through development in *C. elegans*. The manuscript does not only provide very useful descriptive information (albeit without cell-specific resolution) but it also delves deeper into the regulation and function of a highly dimorphic neuropeptide, *ins-39*. I recommend it for publication in Nature Communications but the authors should first address the points I raise below.

- Regarding the preparation of the samples, I understand that it is difficult to get male enriched populations and one needs to use tricks. However, I am concerned about the possibility that some of the differences in gene expression identified may be due to the procedures used to get males. Specifically, could the authors clarify whether hermaphrodites were also arrested at L1, as males were, which may impact on gene expression. And, were hermaphrodites also *him-8*? And if not, why not?
- On the regulation of *ins-39* by *mab-3*: the expression of *ins-39* in masculinised hermaphrodites looks even broader than in males. Please discuss in results a little more whether this is the case and the interpretation of this result. Also, is the ectopic/de-repressed expression of *ins-39* in masculinised

hermaphrodites also dependent on *mab-3*, like its endogenous expression? and similarly in males, is the residual expression of feminised males, *mab-3* dependent? Essentially, where does *mab-3* regulation of *ins-39* sit within the sex determination pathway?

- Please cite “Michael Ailion , James H Thomas, Dauer Formation Induced by High Temperatures in *Caenorhabditis elegans*, *Genetics*, Volume 156, Issue 3, 1 November 2000, Pages 1047–1067,” when mentioning that males are more efficient at dauer entry (in section relating to figure 6E) as this reference already shows so.
- Fig 6G: the text mentions that the difference in L1 survival between wt and *ins-39* mutants is significant but the plot does not look obviously different and there are no stats indicated. Are there statistically significant differences? Then, please show.

REVIEWER COMMENTS

Reviewer #1 (Remarks to the Author):

This study introduces an RNAseq dataset of larval and young adult animals from all developmental stages in both *C. elegans* sexes. While other RNAseq datasets exist from subsets of these conditions, this represents the most comprehensive dataset in *C. elegans*. Additionally, provides solutions to the difficulty in isolating populations of *C. elegans* males, first to generate the RNAseq dataset, and then alternative methods based on what was learned from the dataset itself. This dataset enables the authors to identify dimorphically-expressed genes, such as the insulin-like peptide INS-39. The authors attempt to demonstrate and interpret some functional relevance of this dimorphically-expressed insulin, although the effects seem mild and difficult to interpret (see final comment below). This is the only real weak point in the presentation of a dataset that will surely be of broad interest for the field.

Detailed Comments:

p.13: The claim that *srj-49* is a noncoding RNA is unsubstantiated, and an unlikely explanation based on the data actually provided. If *srj-49* is spliced (even if it is not translated), the SL2::GFP::H2B should be expected to be expressed. If the actual mapped RNAseq reads suggest that *srj-49* transcripts only appear unspliced (ie no evidence of exon/exon junctions at all), then maybe this hypothesis can stand, but the amplification of cDNA in S4B suggests otherwise.

We accept the criticism raised by the reviewer. Our qPCR analysis and DNA electrophoresis of PCR products strongly indicate splicing of the *srj-49* mRNA, as the designed primer spans an exon-exon junction, allowing us to amplify the spliced product. We agree with the reviewer that the suggested hypothesis is not substantiated, and we revised our statement to "*srj-49 could potentially be regulated post-translationally in a sex specific-manner, as we have previously demonstrated for a synaptic receptor*" (Lines 266-267).

Fig 4: Panel B seems to indicate little-to-no expression of INS-39 in the hermaphrodite (the only indicated expression seems to be faint AFD in L1, and then nothing perceptible in any other stage besides maybe something on top of the pharynx in L4). However, panel D shows clear expression in multiple YA hermaphrodite neurons. These two panels use the same endogenous reporter, so they should presumably show the same expression pattern.

Figure 4b compares total fluorescence levels using mean intensity measurements. To do so, we used *low laser intensity* to capture the expression pattern of the *ins-39(syb4915)* reporter, maintaining identical microscope settings between the sexes and throughout all developmental stages. This was necessary because the fluorescence intensity in males is exceptionally high. Attempting to visualize the lowest hermaphrodite expression level would have resulted in a saturated signal in males, which could lead to incorrect comparative analysis. In panel 4d we did increase the laser intensity in hermaphrodites to visualize all expressing neurons. This is now clearly stated in the figure legend. An example of overexposed hermaphrodites across the stages is presented below. After overexposure, you can observe the expression in only 2 neurons in young adult hermaphrodites, as shown in Figure 4d.

Fig 5: With so few expressing neurons, the seemingly weak effect especially in panel H, and the fact that *ins-39* was already combined with NeuroPAL in Fig 4, it would be more satisfying to have neuron IDs rather than simply counting expressing cells in this figure. Also, entitling this figure “spatial localization” seems to imply that *INS-39* is mis-localized by changing sexual identity rather than differentially regulated. A final minor point, the GFP reporter is labeled as *INS-39::GFP* in all images of this figure, where it should presumably be indicated as *SL2::GFP::H2B*.

We agree that assigning cellular IDs to the expressing neurons is insightful. Unfortunately, the panneuronal feminization strain (*ins-39(syb4915[ins-39::SL2::GFP::H2B]); otEx6775 (rab-3::tra-2(ic)::SL2::NLS::tagRFP); otIs525[lim-6(intron4)::gfp]; him-5(e1490) V*) contains a TagRFP co-expressed with the panneuronal feminization construct, rendering it technically challenging to identify the neurons using NeuroPAL. As an alternative approach, we used two strategies to label specific neurons. (1) We identified specific neurons by a Vybrant lipophilic dye stain (DiD) on a strain that also contains an AFD marker, and (2) used a transgenic strain with markers for ASE and AWC (*ceh-36p::tagBFP; flp-6p::BFP*). Using these strains and staining, we found that feminization resulted in no change in expression levels of *INS-39* in ASK, ASJ and AFD but was significantly affected in AWC (and a slighter effect also in ASE). This has been added to the text and figures (Figure 5g and Supplementary Fig. 7, also see below).

As per the reviewer's correct remark, and due to the new results of *mab-3*, we changed the title of the figure to: "*TRA-1 and MAB-3 control INS-39 dimorphic neuronal expression*" (Line 401).

We corrected the labeling in Figure 5 to "*ins-39::SL2::GFP::H2B*".

Supplementary Fig. 7. Pan-neuronal feminization affects *INS-39* expression specifically in AWC, and to a lesser extent in ASE neurons. a. Representative confocal micrographs of a young adult hermaphrodite and male expressing the *INS-39::SL2::GFP::H2B* reporter and co-stained with Vybrant lipophilic dye (DiD), enabling identification of sensory amphid neurons. Scale bars represent 10 μ m. b. Representative confocal micrographs of a young adult hermaphrodite and male expressing the *INS-39::SL2::GFP::H2B* reporter with AFD marker. Scale bar 10 μ m. c. Schematic representation of predicted TRA-1 transcription factor binding site on the exonic region of *ins-39*. Binding-site was determined using RSAT matrix-scan search tool (see methods). d. Representative confocal micrographs of the *ins-39(syb4915)* reporter expression in a wild-type male and pan-neuronally feminized male,

expressing *rab-3p::tra-2(IC)* and co-stained with Vybrant lipophilic dye (DiD), enabling identification of sensory amphid neurons. Co-injection marker is indicated by a yellow asterisk mark. Scale bars represent 10 μ m. e. Representative confocal micrographs of the *ins-39(syb4915)* reporter expression in a wild-type male and pan-neuronally feminized male, expressing *rab-3p::tra-2(IC)* with ASE and AWC (*ceh-36p::tagBFP; flp-6p::BFP*) markers enabling identification of sensory amphid neurons. Co-injection marker is indicated by a yellow asterisk mark. Scale bars represent 10 μ m. f. Quantification of the % of animals expressing INS-39 in the relevant sensory neurons. For calculating statistics, worms were grouped into full expression (both the neurons) and partial expression (0 or 1 neuron), and statistical analysis was calculated using Fisher's exact test. n=14 for both groups, * p < 0.05.

Fig S6: Fluorescence images should be shown for the *dmd-8* and *dmd-9* mutants. This figure and the results are a bit hard to interpret because the expectation seems to be that *dmd-9* is regulating INS-39 only in AFD (maybe AWC?) but the quantification in panels D and E doesn't seem to have been performed in a neuron-specific manner (I was unable to find sufficiently specific details to confirm this in the Methods).

We have now included confocal images for the *dmd-8* and *dmd-9* mutants and their control in both sexes (Supplementary Fig. 8). We carried out a more thorough analysis of INS-39 expression in *dmd-9* mutants in a neuron-specific manner in hermaphrodites and males using neuron-specific markers as described above. While there was a notable global decrease in INS-39 in both sexes in the *dmd-9* mutant, this decrease reached significance only for AFD in hermaphrodites), as speculated by the reviewer (Supplementary Fig. 8b-c). These results suggest that *dmd-9* indeed regulates *ins-39* expression in AFD neurons in hermaphrodites (as expected from the data in Supplementary Fig. 8 panel a, also specific for hermaphrodites), whereas other TFs may be involved in males. We have now revised our text accordingly (line 380-391).

p.24: Figure 4 indicates that INS-39 is not expressed in hermaphrodite ASJ at any stage. Thus, the rationale for invoking ASJ's role in cold tolerance (and the hermaphrodite improved survival rate that is *ins-39*-dependent), is unclear.

Our rationale for carrying out this experiment was based on the high expression of INS-39 in ASJ neurons in males, which led us to the initial hypothesis that its function might be exclusive to males, for example suppressing cold tolerance. As the reviewer correctly pointed out, we found a slight effect only in hermaphrodites. This observation can be attributed to our use of INS-39 knockout, which was not limited exclusively to ASJ neurons. We agree with the reviewer that this assay doesn't add to the conclusions of the manuscript, and we removed it.

Fig. 6G: The text indicates that survival is "significantly longer," but no statistics are shown or indicated in the figure legend or methods.

We apologize for this omission. We have included the detailed statistics under the method sub-heading "L1 survival assay": "*The experiments were combined for analysis and represented using survival graphs in Prism. In the final datasets, the animal count was contingent upon the maximum number of days of surviving animals. The 'n' value for each group in the final datasets was as follows: Wild-type L1 Herm n=2676, INS-39 KO L1 Herm n=1421, Wild-type L1 Male n=2241, INS-39 KO L1 Male n=5453. To assess and compare survival rates, the Kaplan-Meier method was employed. Statistical significance was determined using the Log-rank (Mantel-Cox) test in Prism (GraphPad)*". This data also appears in the figure legend.

p. 25: The conclusions "we suggest that the lower expression levels of INS-39 in hermaphrodites is fine-tuned to critically respond to environmental changes, providing a survival advantage in the face of unfavorable conditions. In contrast, the higher expression in males might hinder its functionality, creating

a ceiling effect,” are a bit confusing. For instance, in the cold tolerance assay, *ins-39* KO has a mild (positive) effect on hermaphrodite survival but none in males. Additionally, *ins-39* KO hermaphrodites enter dauer at a higher rate in response to stress; this could be considered to be an advantage for survival. The claim that the lower expression levels critically respond to environmental changes would suggest that hermaphrodites should always be compromised by loss of *ins-39*? In addition, I would suggest avoiding claims about the functionality of variations in expression level (ie invoking a ceiling effect) when all experiments show all-or-none changes and with conflicting results such as these. These interpretations are also presented in the Discussion section.

We refined our interpretation and rephrased our statement as follows: “Taken together, while in males *ins-39* levels seem mostly unresponsive to the environmental conditions tested in this study, in hermaphrodites *ins-39* modulation plays a role in context-specific response patterns.” (Lines 466-468).

Reviewer #2 (Remarks to the Author):

The authors describe the results of whole animal RNAseq transcriptome analysis for *C. elegans*. They have analyzed RNA samples prepared separately from each of the four developmental stages plus young adults, and separately for each sex (10 averaged samples in all). Transcriptome comparison of the sexes of *C. elegans* has been carried out before. What is new and valuable here is the authors have developed a method to separate males and hermaphrodites at early larval stages. Thus, comparison of the transcriptomes of the two sexes at early larval stages has not been done before. The transcriptome data of Supplementary Table 2 is valuable.

We thank the reviewer for the comments. Our manuscript is novel on multiple fronts. As the reviewer stated, we developed a methodology that enables large-scale isolation of early larval *C. elegans* males with extremely high purity. The comprehensive gene expression atlas for both sexes across development discovered numerous differentially expressed genes, including neuronal gene families like transcription factors, neuropeptides, and GPCRs. We identify thousands of sexually dimorphic genes, among them 1047 new conserved genes associated with human genetic disorders. All our results are openly shared, and since the launch of our website, there have been hundreds of unique entries.

Our approach enabled the identification of an early-stage marker for males, offering another novel tool for the efficient isolation of males in high-throughput experiments. The second part of the paper focuses on one gene that we uncovered, the insulin-like neuropeptide *INS-39*, and offers mechanistic insights into its sexually dimorphic expression and functions. Below we address all the specific comments raised by the reviewer.

A difficulty with this paper is that there is insufficient comparison of these new results with previous data. The authors note a number of limitations of earlier work, but their comparison is inaccurate in one important respect. They state in the introduction and again in the discussion that earlier results were limited by small sample sizes, due to the necessity of, in some cases, picking males from a mixed population by hand. The RNAseq method is so sensitive that small sample size is not a limitation. In fact, Kim et al., (2016), who picked males by hand and thus have a pure separation of the sexes, detected expression of more genes (21,143, including 17,967 protein coding) than the present work (14,185). This is particularly surprising since Kim et al did not include data from L1 or L2 larval stages.

While we recognize the significant contributions made by *Kim et al.*, our work takes a significant step further by focusing on a few crucial aspects that were previously unexplored: the stage-specific transcriptomics of both sexes and conducting cross-analysis between them. A significant contribution is the incorporation of the early larval transcriptome. In Kim et al, samples (L3-adults) for each sex were

merged for analysis, and in addition, only one sample from each stage was collected. When referring to "small sample size" in the introduction, we referred to few technical or biological sample replicates. We have now clarified this in the manuscript by using the term "few stage-specific sample replicates". We recognize that the phrase "small sample sizes" was confusing.

In the discussion, we intended to convey that the low number of identified TFs in the previous studies is likely due to the pooling of all the samples from different developmental stages. We removed the sentence from the discussion.

With regards to gene count, our RNA-seq indeed identified less genes (14,185) than a prior study (21,143, including 17,967 protein coding). However, we feel it is not scientifically accurate to compare the robustness of the sequencing method and its quality based on the total number of genes, as the difference in gene number could be explained by several differences in technical features of both studies listed below:

1. Technical variability, such as differences in library preparation methods. We prepared our library using MARS-seq protocol. We note that we are unable to access Kim et al library preparation protocol via their published link http://wasp.einstein.yu.edu/index.php/Main_Page. MARS-seq is a 3' end library preparation protocol that excludes the majority of the non-protein coding genes. This may partly accounts for the higher gene count detected in *kim et al, 2016*.
2. MARS-seq had a substantially lower sensitivity but has higher true positive rates when compared to other RNAseq library preparation protocol (<https://pubmed.ncbi.nlm.nih.gov/28212749/>). We now note this limitation in the discussion (Lines 501-505).

When comparing the protein-coding genes, several factors are responsible for the differences:

3. Bioinformatics pipelines: We applied strict cutoffs by removing (trimming) adapters, polyA/T sequences, and low-quality bases at the 3' end. Reads shorter than 25 bases were discarded by the analysis process using cutadapt. (DOI:10.14806/ej.17.1.200) (parameters: -a AGATCGGAAGAGCACACGTCTGAACTCCAGTCAC -a "A{10}" -times 2 -u 3 -u -3 -q 20 -m 25).

We applied strict criteria for keeping only reads that mapped uniquely to genes using STAR.

On average, we removed 37% of the reads during the entire analysis process up to the read count stage (see graph below), thus reducing much of the background noise. It's important to note that *Kim et al* didn't use the same methodology.

4. Discrepancies in gene annotations between the two studies can influence the count of identified genes. In our study, we used an updated annotation database (WBcel235) compared to the previously used database (WS190).
5. Random Sampling: RNA-seq is a stochastic process, and the random sampling of RNA molecules during library preparation and sequencing can lead to some variability in the genes detected, especially for genes with low expression levels.

We have now revised our introduction and discussion sections to provide a more accurate and balanced comparison with Kim et al.'s work.

Considering what is new here, transcriptomes of L1 and L2 larvae, comparison of the sexes at early larval stages should have been emphasized in the analysis. While the authors do describe their results, they highlight no particular significant findings. Rather, they focus on male-specific and nervous-system specific expression. Since this is precisely the question also analyzed by Kim et al (2016), the present authors need to justify this choice. A comparison of the gene lists should have been included.

Our perspective on the novelty of the work, as elaborated also in the previous comment, is somewhat different. Throughout the first part of the manuscript, which describes our transcriptome analysis, we present data for DEGs and gene families across all developmental stages and in the two sexes, with their appropriate significance values and log fold change (Figure 1, Figure 2, Figure 3, Supplementary Fig. 6). This type of analysis **between developmental stages within each sex** is, to the best of our knowledge, novel. Starting from Figure 1, we discuss the results of the gene ontology enrichment analyses across development in the two sexes (lines 160-167). This analysis highlights the gene families that are enriched at each developmental stage, in an unbiased manner, not focusing on the nervous system, as suggested by the reviewer. We have extended this section in the text to highlight significant findings and also included a new table that summarizes the L1-L2 DEGs (Supplementary Table 5).

We focus on a few genes that emerged from our results-218 transcription factors were differentially expressed in at least one developmental stage (Supplementary table 6). Figure 2 shows the top 25 TFs, two of which are differentially expressed at L1-L2, *ces-2* and *mab-3*. *ces-2* has been further characterized in the manuscript (Figure 2C, Supplementary Fig. 5E). For *mab-3*, we now report a novel **hermaphrodite-specific neuronal function** in the AFD neuron (Figure 5). Additionally, Figure 3 shows early differential expression of GPCRs and several neuropeptides.

As per the reviewer's suggestion, we added a comparative analysis of our dataset and Kim et al's. Specifically, we compared our dataset, which included 4,525 genes exhibiting significance in either the L3, L4, or YA stages, with Kim et al.'s male and hermaphrodite enriched datasets (as depicted in the figure below). Comparing the two datasets we noticed that, within the shared gene list, certain genes exhibit stage-specific expression. Just to give a few examples, although the genes *ceh-60*, *nspd-3*, *smz-1*, *snf-2* and *snf-4* are common to both studies, our study provides crucial temporal sex-specific gene expression dynamics. The developmental plot of *smz-1* below shows the expression dynamics in both sexes over time with significance values.

Data set comparison: a. Venn diagram comparing differentially expressed genes in both the sexes in this study with *kim et al*, 2016. b. Bubble plot representation of *ceh-60*, *nspd-3*, *smz-1*, *snf-2* and *snf-4* genes differentially expressed in any of the five developmental stages of the two sexes. Bubble size represents \log_2 of fold change in expression of that gene (only genes that passed the filter $\text{padj} \leq 0.05$, $|\log_2 \text{fold change}| \geq 1$ and $\text{basemean} \geq 5$ are plotted), and bubble color represents enrichment in either sex (male enrichment in cyan, hermaphrodite enrichment in orange). Error bars are standard error of the mean (SEM). c. Normalized expression values of *smz-1* across the experimental stages in both sexes from this manuscript and *kim et al.*, 2016 (cyan: males, orange: hermaphrodites). Adjusted p-values were calculated by Wald test for each comparison performed by DESeq2, * $p < 0.05$.

Was anything new found? A few spot checks in fact uncovers discrepancies—Kim et al, for example, do not show *ceh-13* or *nob-1* as male-enriched transcription factors.

The novelty issue was rigorously addressed in the previous comments. To address the comment on discrepancies between the data sets we performed qPCR analysis on *nob-1*. The analysis demonstrates a significant correlation with our RNAseq datasets, validating that *nob-1* is enriched among male-specific transcription factors during the YA stage. A previous publication has already shown male-specific *ceh-13* expression in the sensory rays during the YA stage ([https://doi.org/10.1016/S0012-1606\(03\)00138-6](https://doi.org/10.1016/S0012-1606(03)00138-6)).

Figure: Real-time qPCR analysis of *nob-1* and *ceh-13* mRNA expression in YA hermaphrodites and males normalized to housekeeping gene (*pmp-3*). n=3 biological repeats. We performed a Welch's t-test. ** p < 0.01.

A spot check finds a discrepancy within this manuscript itself: *ttx-1* is described as expressed higher in males during sexual maturation, but unless this reviewer misunderstands the data, expression of *ttx-1* is in fact higher in L1 males (201 average normalized reads) than in L3 males (39 average normalized reads) or L4 males (43 average normalized reads).

Our statement wasn't phrased clearly enough, we meant to say that the expression was significantly different between the two sexes during sexual maturation. Indeed, as the reviewer noticed, *ttx-1* is much higher during L1 stage in both sexes. We now corrected the statement in line 227 so that the meaning is clear. Additionally, we corroborated this finding by qPCR analysis of *ttx-1* (see below). We also define the timing of "sexual maturation" (line 58-59).

Real-time qPCR analysis of *ttx-1* mRNA expression in YA hermaphrodites and males normalized to a housekeeping gene (*pmp-3*). n=3 biological repeats. We performed a Welch's t-test. * p < 0.05.

For the second part of the paper, the authors select one gene enriched in male expression, *ins-39*, for further study. But they don't learn anything in particular about it and there is no justification for this direction of the study. They do not learn the mechanistic basis of its male-specific expression or the functional reason the gene is expressed in a few male-specific neurons.

We would first like to comment concerning the last statement that *ins-39* is NOT expressed in male-specific neurons, but rather in sex-shared neurons and in surprisingly higher levels. RNAseq data reports an 8.80361 log2 fold change for male-biased expression (x446.8386 change). This intriguing rise in expression levels in sex-shared neurons is what led us to pursue its possible functions in males. Our investigations led to several discoveries listed below:

1. *INS-39* emerged as a valuable early-stage marker for males, providing a previously unavailable tool for efficiently isolating males in high-throughput experiments.
2. We carried out a comprehensive examination of its anatomical localization and showed that its expression is restricted to 2 sex-shared neurons, AFDs and ASKs in hermaphrodites, and 3 additional pairs of sex-shared sensory neurons in males. Sex reversal experiments revealed that in hermaphrodites *TRA-1* suppresses *ins-39* expression to AFDs only (an intriguing result on its own), but its forced expression in males is not sufficient to drive *ins-39* expression, suggesting that additional factors are required for its dimorphic male expression. Lastly, we found a few *dmds* that regulate *ins-39* expression- we describe a novel role for *mab-3* in **hermaphrodites**, where it is required for the maintenance of *ins-39* expression in adults. Adding a new epistasis experiment we now show that *mab-3* loss of function suppresses the masculinization phenotype, supporting its central role in regulating *ins-39* expression levels (Figure 5b-c, lines 373-379). In males, *mab-3* controls *ins-39* expression, and along with additional factors functions to promote the extremely high *ins-39* levels in

the sensory neurons (Figure 5g-h). Additional data we added suggests at least one of these additional factors is *dmd-9* (Supplementary Fig. 8 b-c).

3. As for the functional role of *ins-39*, we report several new findings summarized below:
 - I. Corroborating a recent study (<https://elifesciences.org/articles/78941>), we also show that INS-39 expression in the hermaphrodite AFD neurons responds to temperature shifts. The lack of response to temperature shifts in *ins-39* mutant males is interesting and aligns with the lack of effect of males in peroxide resistance assays. Negative results, especially when appearing in a dimorphic context (single-sex *ins-39* dependent functions) are also important and insightful. Here, we speculate that the low levels in hermaphrodites allow *ins-39* to serve as an environmental sensor, whereas in males its high levels suppress these abilities. We also speculate that additional insulin-like peptides might play male-specific roles, which remain to be discovered.
 - II. We report a novel role for *ins-39* in dauer entry during early development. *ins-39* mutants hermaphrodites were more efficient in entering the dauer stage.
 - III. Lastly, we report sex shared role of *ins-39* in L1 survival. *ins-39* KO L1 animals of both sexes survived significantly longer than their wild-type controls with a more pronounced role in males.

Taken together, the emerging picture is that of *ins-39* having pleiotropic roles in hermaphrodites in the behavioral readout we examined. We refined our conclusion in the last result paragraph it now reads: *Taken together, while in males ins-39 levels seem mostly unresponsive to the environmental conditions tested in this study, in hermaphrodites ins-39 modulation plays a role in context-specific response patterns.* (Lines 466-468). We also acknowledge the lack of *ins-39*-dependent behaviors in the discussion: *“Our dauer entry and peroxide resistance assays show that for hermaphrodites the absence of ins-39 improves their survivability, while it has no effect on males. The tight regulation of ins-39 expression levels in hermaphrodites suggests that maintaining low levels is critical for the animal. Why males require high ins-39 levels is unknown but may represent an evolutionary cost that comes as a tradeoff for an unknown advantage that awaits further research”.* (Lines 534-539). In line with our manuscript being a rich source, we have opened several avenues for further research, one of which would be deciphering the precise role of *ins-39* in males.

Reviewer #3 (Remarks to the Author):

This is a very comprehensive analysis of sexually dimorphic gene expression through development in *C. elegans*. The manuscript does not only provide very useful descriptive information (albeit without cell-specific resolution) but it also delves deeper into the regulation and function of a highly dimorphic neuropeptide, *ins-39*. I recommend it for publication in Nature Communications but the authors should first address the points I raise below.

Regarding the preparation of the samples, I understand that it is difficult to get male enriched populations and one needs to use tricks. However, I am concerned about the possibility that some of the differences in gene expression identified may be due to the procedures used to get males. Specifically, could the authors clarify whether hermaphrodites were also arrested at L1, as males were, which may impact on gene expression. And, were hermaphrodites also *him-8*? And if not, why not?

We thank the reviewer for his/her feedback. The hermaphrodites used in our study underwent L1 arrest identical to that of males. Analysis for both sexes was carried out in identical procedures during our analysis. To enhance clarity, we have included the phrase "using an identical isolation protocol" in the 'RNA Isolation and Library Preparation' subsection of the methods. The hermaphrodites used in our study were Bristol N2 wildtype strain. Utilization of a *him-8* mutant strain results in a high incidence of males,

constituting approximately 30-40% of the population. This is the reasoning for not using *him-8* hermaphrodites.

The *dpy-28* protocol was also successfully used by us to generate a single-cell atlas of the male and hermaphrodite nervous system (Haque et al., manuscript in preparation). As can be seen below, male and hermaphrodite single neurons largely overlap, with specific DEGs that we further validated in that study. Our male data is highly similar to that obtained by the CenGEN consortium, which utilized another method for male neuron purification (personal communication). Thus, we are highly confident that the male purification method is reliable and robust.

Single-cell RNA sequencing in both sexes of *C. elegans* A) UMAP representation of all cell clusters identified with their neuronal identity. We identified sex-specific expression patterns of over 10,555 genes spanning 112 distinctly identifiable neuronal cell types in the adult nervous system. B) UMAP representation of cell clusters identified segregated into cells from YA hermaphrodites (red), cells from YA males (green), and cells from L4 hermaphrodites CenGEN dataset (yellow). The subtle shift in male and hermaphrodite datasets with the previously identified L4 hermaphrodites' clusters indicate that they have similar but not totally overlapping transcriptome profiles. C) Receiver-Operator Characteristic (ROC) curve of True Positive Rate (TPR) versus False Positive Rate (FPR) for hermaphrodites, males, hermaphrodites-males together, and CenGEN compared to ground truth expression data. Increased stringency diminishes both the TPR and FPR. (Haque et al., Manuscript in preparation).

On the regulation of *ins-39* by *mab-3*: the expression of *ins-39* in masculinised hermaphrodites looks even broader than in males. Please discuss in results a little more whether this is the case and the interpretation of this result.

Also, is the ectopic/de-repressed expression of *ins-39* in masculinised hermaphrodites also dependent on *mab-3*, like its endogenous expression? and similarly in males, is the residual expression of feminised males, *mab-3* dependent? Essentially, where does *mab-3* regulation of *ins-39* sit within the sex determination pathway?

We thank the reviewer for this comment as it greatly improved our understanding of *ins-39* regulation. As per the reviewer's suggestion, we conducted an epistasis analysis between *mab-3* mutants and feminized or masculinized males and hermaphrodites (respectively). Our results nicely show that *mab-3* loss of function suppresses the ectopic expression observed in masculinized hermaphrodites. Thus, *mab-3* acts downstream to *tra-1* to control *INS-39* (Figure 5b, g). Since the sex-reversal phenotype was not fully suppressed by *mab-3* mutations, an additional, *tra-1*-independent pathway exists. The expression of *INS-39* is regulated by many pathways in a cell-autonomous way, which can be either *tra-1* or *tra-1* and *mab-3* dependent, or they can be an unidentified system that functions independently of both. We now discuss this in greater detail in the results section considering the latest findings (line 364-399).

Please cite "Michael Ailion, James H Thomas, Dauer Formation Induced by High Temperatures in *Caenorhabditis elegans*, *Genetics*, Volume 156, Issue 3, 1 November 2000, Pages 1047–1067," when

mentioning that males are more efficient at dauer entry (in section relating to figure 6E) as this reference already shows so.

We thank the reviewer for pointing this out. We have now included the suggested reference in our manuscript. The modified text now reads “We first tested the ability of the animals to enter the dauer stage and found that wild-type males were more efficient in dauer transition than hermaphrodites, consistent with previous findings (Figure 6e)” (Lines 452-454).

Fig 6G: the text mentions that the difference in L1 survival between wt and ins-39 mutants is significant but the plot does not look obviously different and there are no stats indicated. Are there statistically significant differences? Then, please show.

We have included the detailed statistics under method sub-heading “L1 survival assay”. The included text is “*The experiments were combined for analysis and represented using survival graphs in Prism. In the final datasets, the animal count was contingent upon the maximum number of days of surviving animals. The 'n' value for each group in the final datasets was as follows: Wild-type L1 Herm n=2676, INS-39 KO L1 Herm n=1421, Wild-type L1 Male n=2241, INS-39 KO L1 Male n=5453. To assess and compare survival rates, the Kaplan-Meier method was employed. Statistical significance was determined using the Log-rank (Mantel-Cox) test in Prism (GraphPad)*”.

REVIEWERS' COMMENTS

Reviewer #1 (Remarks to the Author):

The authors have satisfactorily responded to all of my concerns, and the revised manuscript is much improved and suitable for publication.

Reviewer #2 (Remarks to the Author):

I do not find this revised version to be substantially changed compared to the original. For the first part of the paper, the explanations for the differences in the total number of genes detected as compared to Kim et al, based on the different methodologies, are unsatisfying. I don't understand the statement that Kim et al merged their data from different stages. This is not seemingly correct, depending on what they mean. The stages were treated separately. The Venn diagram comparing this new data with Kim et al is disturbing. The authors examine a few discrepancies and justify their result. But what are we to make of the hundreds of others?

Regarding the second part of the paper, on ins-39, it remains the case that selecting just one differentially expressed gene and studying it seems neither here nor there. And it remains true that mostly the mechanism and functional significance of the differential expression remains a mystery.

Reviewer #3 (Remarks to the Author):

The authors have addressed my initial concerns satisfactorily and the manuscript is much improved. I am also satisfied with their response to the comments raised by the other two reviewers.